# Chicks4FreeID: A Benchmark Dataset for Chicken Re-Identification

**Daria Kern**[1,2] *   **Tobias Schiele**[1,2] *   **Ulrich Klauck**[1,3]   **Winfred Ingabire**[2]

[1]Aalen University, Germany
{daria.kern, tobias.schiele, ulrich.klauck}@hs-aalen.de
[2]Glasgow Caledonian University, United Kingdom
winfred.ingabire@gcu.ac.uk
[3]University of the Western Cape, South Africa

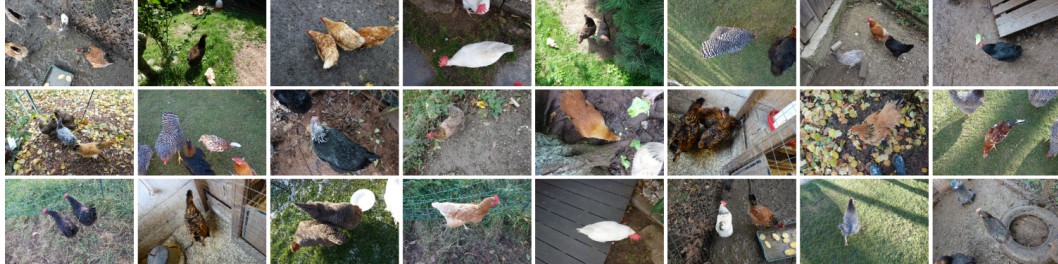

Figure 1: Excerpt from the Chicks4FreeID dataset.

## Abstract

To address the need for well-annotated datasets in the field of animal re-identification, and particularly to close the existing gap for chickens, we introduce the Chicks4FreeID dataset. This dataset is the first publicly available re-identification resource dedicated to the most farmed animal in the world. It includes top-down view images of individually segmented and annotated chickens, along with preprocessed cut-out crops of the instances. The dataset comprises 1215 annotations of 50 unique chicken individuals, as well as a total of 55 annotations of 2 roosters and 2 ducks. In addition to re-identification, the dataset supports semantic and instance segmentation tasks by providing corresponding masks. Curation and annotation were performed manually, ensuring high-quality, nearly pixel-perfect masks and accurate ground truth assignment of the individuals using expert knowledge. Additionally, we provide context by offering a comprehensive overview of existing datasets for animal re-identification. To facilitate comparability, we establish a baseline for the re-identification task testing different approaches. Performance is evaluated based on mAP, Top-1, and Top-5 accuracy metrics. Both the data and code are publicly shared under a CC BY 4.0 license, promoting accessibility and further research. The dataset can be accessed at https://huggingface.co/datasets/dariakern/Chicks4FreeID and the code at https://github.com/DariaKern/Chicks4FreeID.

---

*contributed equally. Contact: Chicks4FreeID@dariakern.com

Submitted to the 38th Conference on Neural Information Processing Systems (NeurIPS 2024) Track on Datasets and Benchmarks. Do not distribute.

# 1 Introduction

## 1.1 Motivation

Chickens struggle to recognize other individuals after visible changes are applied to the comb or plumage [1]. Much like chickens are able to use visual cues to differentiate each other, artificial intelligence (AI) is capable of utilizing image or video inputs for re-identification purposes. AI-driven re-identification and tracking systems hold great potential for enhancing animal husbandry and livestock farming. These systems may allow for the observation of social structures and behavior, enhance welfare, and potentially lead to more efficient animal management with minimal disruption to the livestock [2]. They also may help assess health and well-being, i.e., by providing crucial traceability during disease outbreaks. Furthermore, they offer a cost-effective and non-invasive alternative to manual tagging methods.

Despite the significant potential, there is a notable gap in publicly available datasets for such technologies, especially for chickens — the most farmed animal globally. Remarkably, to our knowledge, no publicly available dataset for chicken re-identification exists, highlighting an urgent need for development in this field. Public datasets for the task of individual animal re-identification in general are scarce [3, 2]. In particular, well-annotated datasets [4]. The practice of openly sharing data and code should be encouraged to enhance result comparability, yet not all research data are currently made public. In their work, [5] emphasize the importance of creating and sharing publicly available and well-annotated benchmark datasets for the task of animal re-identification.

Establishing a benchmark dataset involves evaluating how well existing methods solve the dataset. The reported metrics serve as a baseline for future researchers to report their improvements. Given the diverse nature of research, it is important for the baseline to cover common approaches and common metrics. This ensures that the achievements of future researchers can be effectively compared, facilitating a standardized assessment of advancements in the field.

## 1.2 Contribution

We address the existing gap and present our Chicks4FreeID dataset, which does not only support the task of re-identification but also semantic and instance segmentation. We make this thoroughly documented dataset freely accessible to the research community and the public. The dataset includes 54 individuals, of which 50 are chickens. Each occurrence is nearly pixel-perfectly segmented, resulting in 1270 instance masks. Based on the cut-out crops of 1215 chicken instance masks, we provide an initial baseline for the task of closed set re-identification. This allows the research community to compare their methods and results effectively. In summary:

    i We provide a comprehensive overview of publicly available datasets for animal re-identification.

    ii We introduce the first publicly available dataset for chicken re-identification.

    iii We establish a baseline for closed set re-identification on the introduced dataset.

# 2 Related work

## 2.1 Animal re-identification

Animal re-identification, the task of identifying individual animals within one (or sometimes several) species, finds applications in various fields. Particularly in wildlife conservation efforts, where monitoring endangered species is crucial [6–9]. But also in livestock management, notably cattle [10–14] and yak [15]. Honeybees [4] and bumblebees [16, 17] have also been subject to investigation.

Re-identification falls into one of two categories: closed set and open set. In closed set re-identification, all individuals are known from the beginning, and those to be identified can be matched with identities of a predefined set. In open set re-identification, the identity of the individual

in question may not necessarily be part of a predefined set. It is possible to encounter completely new, undocumented individuals. Such individuals must be annotated as a new identity and, upon subsequent encounters, accurately matched.

While facial recognition is a prevalent method for re-identifying humans [18], the faces of animals can likewise serve as a means to re-identify individuals, as has previously been demonstrated for rhesus macaque [19], chimpanzee [20], cats [21], lions [22], dogs [23], giant pandas [8] and red pandas [9]. However, animals frequently exhibit more distinctive visual traits beyond their faces. For example, natural markings such as stripes [24–27] and scale patterns [28] can serve as prominent identifiers. But also specific body parts can contribute to distinguishing individuals, such as the fins of dolphins [29] and sharks [30]. Similarly to how fingerprints differentiate humans, nose prints of dogs have been utilized to uniquely identify individual dogs [31]. Conversely, little inter-individual variability poses a challenge to the re-identification task. Species exhibiting minimal or subtle visual distinctions between individuals are, for instance, (polar) bears [32, 33] or elephants [34]. Visual traits play a pivotal role in animal re-identification within computer vision, serving as essential markers for distinguishing individuals. However, the task is complex and extends beyond mere visual cues. Factors such as lighting, perspective, body changes over time, and partially obscured body parts pose additional challenges [5].

To further advance the field and aid the research community, [35] released the WildlifeDatasets toolkit - an open-source toolkit for animal re-identification. It gathers publicly available animal re-identification datasets in one place, in an effort to make them more easily accessible and to improve usability. Included are various tools, i.e., for data handling and processing, algorithms relevant to the task of re-identification, pretrained models, as well as evaluation methods. Therewith, they address the prevailing absence of standardization across the literature and facilitate comparability and reproducibility of results. Within their work, they also introduce a new state-of-the-art, the MegaDescriptor, notably the first foundation model for animal re-identification. Likewise, [36] present an open-source re-identification method initially developed for sea stars, which was successfully extended to seven mammalian species without adjustments. They also report state-of-the-art results. Moreover, [37] introduced Tri-AI, a system designed for the rapid detection, identification, and tracking of individuals from a wide range of primate species. The system is capable of processing both video footage and still images. The task of re-identification is closely related to tracking, where individuals are detected and tracked across various video frames. During tracking, individuals often need to be re-identified after leaving and re-entering the field of vision.

## 2.2 Re-identification datasets

A review of existing resources revealed fewer than 40 publicly available datasets for animal re-identification. This leads to the conclusion that a significant number of animal species are not yet covered, including chickens. Birds in general seem to be underrepresented in this domain, with only a couple of datasets available [38, 39]. In fact, a noticeable focus lies on marine life [40–50]. However, cattle are the most frequently featured species [11, 51–55], with much of the data collected by the same group of researchers.

Table 1 provides a summary of the publicly accessible datasets found, arranged by year. Each entry details the name of the dataset ("Dataset"), the associated publication ("Publ."), and species focus ("Species"). "IDs" denotes the number of unique identities present within the dataset. Additionally, the total number of annotated animal instances within all images of each dataset is noted ("Annot."). An indication(*) of whether the data was derived from video sources is given as well. For ease of access, a direct link to each dataset is provided ("Avail. at"). Although all of the datasets are publicly accessible, some are released under licenses that are relatively restrictive.

Table 1: Publicly available animal re-identification datasets, arranged by date of publication. An asterisk (*) marks data derived from video footage.

| Year | Publ. | Dataset | IDs | Species | Annot. | Avail. at |
|---|---|---|---|---|---|---|
| | ours | Chicks4FreeID | 50, 2, 2 | chicken, duck, rooster | 1215, 40, 15 | [56] |
| 2024 | [28] | SeaTurtleID2022 | 438 | sea turtle | 8729 | [40] |
| 2023 | [3] | Mammal Club (IISD) | 218 | 11 terrestrial mammal species* | 33612 | [57] |
| 2023 | [58] | Multi-pose dog dataset | 192 | dog | 1657 | [59] |
| 2023 | [32] | PolarBearVidID | 13 | polar bear* | 138363 | [60] |
| 2023 | [36] | Sea Star Re-ID | 39, 56 | common starfish, Australian cushion star | 1204, 983 | [41] |
| 2022 | [61] | Animal-Identification-from-Video | 58, 26, 9 | pigeon*, pig*, Koi fish* | 12671, 6184, 1635 | [39] |
| 2022 | n.a. | Beluga ID | 788 | beluga whale | 5902 | [42] |
| 2022 | n.a. | Happywhale | 15587 | 30 different species of whales and dolphins | 51033 | [43] |
| 2022 | n.a. | Hyiena ID | 256 | spotted hyena | 3129 | [62] |
| 2022 | n.a. | Leopard ID | 430 | African leopard | 6805 | [63] |
| 2022 | [64] | SealID | 57 | Saimaa ringed seal | 2080 | [44] |
| 2022 | [65] | SeaTurtleIDHeads | 400 | sea turtle | 7774 | [45] |
| 2022 | n.a. | Turtle Recall | 100 | sea turtle | 2145 | [46] |
| 2021 | [66] | Cow Dataset | 13 | cow | 3772 | [11] |
| 2021 | [13] | Cows2021 | 182 | Holstein-Friesian cattle* | 13784 | [51] |
| 2021 | [67] | Giraffe Dataset | 62 | giraffe | 624 | [68] |
| 2021 | [8] | iPanda-50 | 50 | giant panda | 6874 | [69] |
| 2020 | [26] | AAU Zebrafish Dataset | 6 | zebrafish* | 6672 | [70] |
| 2020 | [37] | Animal Face Dataset | 1040 | 41 primate species | 102399 | [71] |
| 2020 | [24] | ATRW | 92 | Amur tiger* | 3649 | [72] |
| 2020 | [73] | Lion Face Dataset | 94 | lion | 740 | [22] |
| 2020 | [74] | NDD20 | 44, 82 | bottlenose and white-beaked dolphin, white-beaked dolphin (underwater)* | 2201, 2201 | [47] |
| 2020 | [73] | Nyala Data | 237 | nyala | 1942 | [75] |
| 2020 | [14] | OpenCows2020 | 46 | Holstein-Friesian cattle* | 4736 | [52] |
| 2019 | [76] | Bird individualID | 30, 10 ,10 | sociable weaver, great tit, zebra finch | 51934 | [38] |
| 2019 | [23] | Dog Face Dataset | 1393 | dog | 8363 | [77] |
| 2018 | [21] | Cat Individual Images | 518 | cat | 13536 | [78] |
| 2018 | [79] | Fruit Fly Dataset | 60 | fruit fly* | 2592000 | [80] |
| 2018 | n.a. | HumpbackWhaleID | 5004 | humpback whale | 15697 | [48] |
| 2018 | [19] | MacaqueFaces | 34 | rhesus macaque* | 6280 | [81] |
| 2017 | [12] | AerialCattle2017 | 23 | Holstein-Friesian cattle* | 46340 | [53] |
| 2017 | [12] | FriesianCattle2017 | 89 | Holstein-Friesian cattle* | 940 | [54] |
| 2017 | [25] | GZGC | 2056 | plains zebra and Masai giraffe | 6925 | [82] |
| 2016 | [20] | C-Tai | 78 | chimpanzee | 5078 | [83] |
| 2016 | [20] | C-Zoo | 24 | chimpanzee | 2109 | [83] |
| 2016 | [10] | FriesianCattle2015 | 40 | Holstein-Friesian cattle* | 377 | [55] |
| 2015 | n.a. | Right Whale Recognition | 447 | North Atlantic right whale | 4544 | [49] |
| 2011 | [27] | StripeSpotter | 45 | plains and Grevy's zebra | 820 | [27] |
| 2009 | [84] | Whale Shark ID | 543 | whale shark | 7693 | [50] |

## 3 The Chicks4FreeID dataset

### 3.1 Data

The Chicks4FreeID dataset contains top-down view images of individually segmented and annotated chickens, with some images also featuring roosters and ducks. Each image is accompanied by a color-coded semantic segmentation mask that classifies pixel values by animal category (chicken, rooster, duck) and background, as well as binary segmentation mask(s) for the animal instance(s) depicted. Additionally, the dataset includes preprocessed cut-out crops (detailed in Section 3.5) of the respective animal instances. Figure 2 gives a first overview of the dataset.

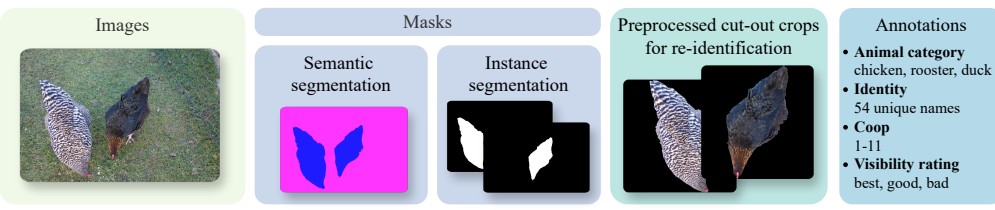

Figure 2: Dataset overview.

## 3.2 Collection

Various coops of private households were visited to photograph chickens. Among these coops, two additionally accommodate a rooster each, while another houses two ducks. A total of 677 images were captured using two similar models of cameras: the "Sony CyberShot DSC-RX100 VI" and the "Sony CyberShot DSC-RX100 I". The resolution of the images stands at 3648x5472 pixels. Every image includes at least one chicken, ensuring no images without chickens are part of the dataset. It was collected over the span of one year, however all images of a coop were shot within one day. In other words, all photos of a given individual were taken on the same day. The images were captured from a top-down view perspective, aiming to capture the plumage. The dataset is not limited to a single breed of chicken, ensuring a certain level of variability.

## 3.3 Annotation

We utilized Labelbox [85] under a free educational license for manual data annotation.

**Instances and background**    For each animal instance appearing within an image, a segmentation was meticulously hand-crafted by a human annotator. No AI has been used during the annotation process to ensure high-quality, nearly pixel-perfect instance masks. The instance masks include the comb, head, beak, and plumage. Feet were excluded as rings could give away the identity. Feet and scattered feathers are considered part of the background, along with any visible objects or living beings that are not chickens, roosters, or ducks. Compared to conventional bounding boxes, instance masks offer the advantage of better supporting the subsequent re-identification process. The background can be easily removed as it might contain unwanted clues about the identity of the chickens. Furthermore, the provided masks render the dataset well-suited for instance segmentation tasks as well.

**Animal categories**    Each instance of an animal was assigned to one of three animal categories. These are "chicken", "rooster", and "duck". Roosters and especially ducks serve as exceptions within the predominantly chicken-based collection. This characteristic potentially positions the dataset as a resource for anomaly detection as well.

**Identities and coops**    The identities of the subjects were meticulously studied prior to photography, closely monitored throughout the image capture process, and ultimately assigned by a human annotator. The ground truth annotation was performed without the use of any algorithm. In cases where the human annotator could not assign an identity, the instance was labeled as identity "Unknown". It is essential to clarify that the label "Unknown" does not imply the presence of a new individual. Instead, it represents an unidentified individual from the closed set, more precisely, from the annotated coop. Each image contains one or more chickens, all of which are individually identified by their unique names. Roosters and ducks are each also uniquely named. Furthermore, each instance is explicitly annotated to indicate the specific coop to which it belongs.

**Visibility rating**    Acknowledging varying visibility of the subjects (chickens, roosters, ducks) within the images, each appearance has been manually assigned a visibility rating, categorized as either "bad", "good", or "best". The "best" rating includes segmentation instances that fully display the subject from the desired top-down perspective, and those where only an insignificant part is missing, such as the very tip of the tail feathers. Instances that include only small parts of the subject and on which the subject is difficult to recognize fall under the "bad" rating. All remaining segmentation instances, that do not qualify as "bad" or "best", are rated as "good".

## 3.4 Composition

The dataset comprises a collection of 677 images, featuring a total of 50 distinct chicken, 2 rooster, and 2 duck identities distributed across 11 different coops. A total of 1270 instances were obtained by segmenting 1215 appearances (instances) of chickens, alongside 15 roosters and 40 ducks.

Each instance is of a certain animal category ("chicken", "rooster", "duck") and was assigned the corresponding coop (1-11), visibility ("best", "good", "bad") and identity (1 of 54 names or "Unknown"). It is important to mention that no "Unknown" instances are present in the "best" or "good" subset. The ground truth identity for all instances in these subsets is, therefore, known. Figure 3 illustrates the number of instances for each individual, as well as the visibility rating of the instances. It starts with the individual with the most instances in the "best" subset and is arranged in descending order. The most represented chicken in the "best" subset is Mirmir with 27 instances, whereas Isolde is the least represented chicken with 4 instances.

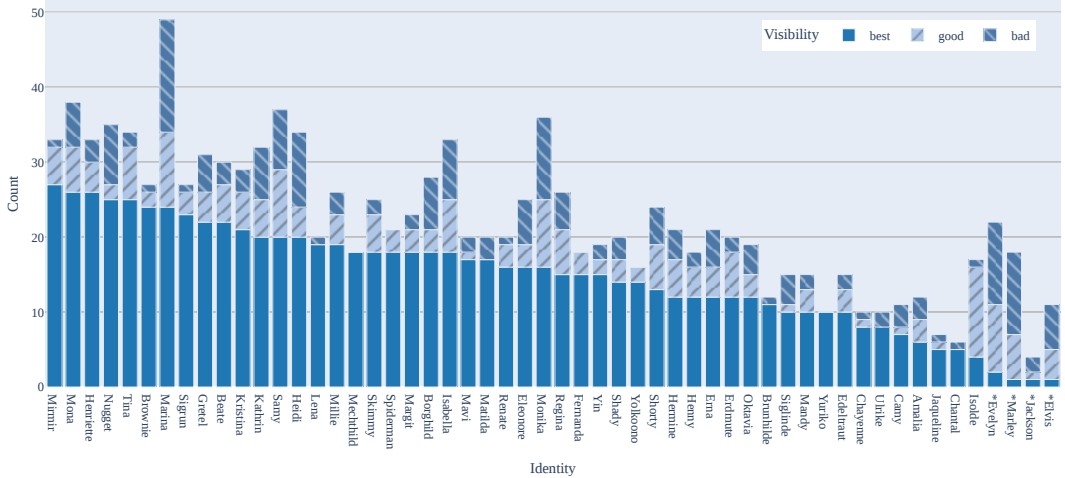

Figure 3: Visibility distributions for all instances of each individual. Ducks and roosters are marked with an asterisk (*).

## 3.5 Preprocessing

The following steps describe the preprocessing procedure to obtain the cut-out crops for the re-identification task. For all individuals captured in an image, a bounding box is created based on the instance masks. In the first step, both the image and the mask are cropped (to the area of interest contained in the bounding box) to focus solely on the individual (see Figure 4: Step 1). The cropped mask is then used to remove the background from the cropped image (Step 2). Finally, the resulting image is adjusted to a square shape for ease of use and consistency (Step 3). The resulting resolutions remain as is, with no resizing taking place.

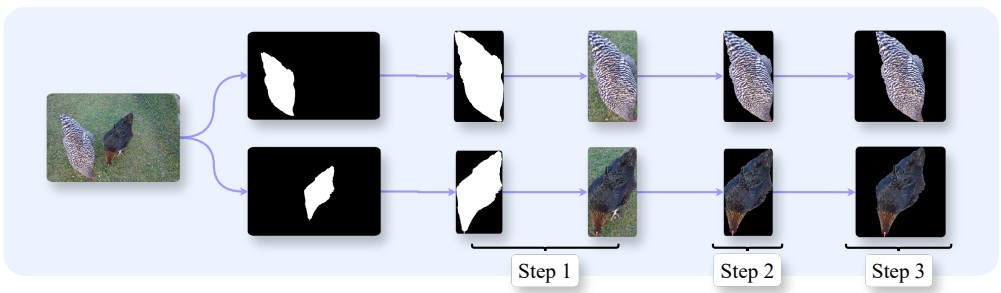

Figure 4: Data preprocessing pipeline for subsequent re-identification.

# 4 Experiments

## 4.1 Dataset, split and augmentation

For the closed set re-identification experiments, we utilize preprocessed cut-out crops as described in Section 3.5. To focus solely on all 50 chicken identities, the four identities of ducks and roosters were excluded. By removing instances of visibility level "good" and "bad", we ensure that only instances with "best" visibility are included. The utilized "best" subset does not contain any "Unknown" instances. The number of chicken instances contained in the "best" subset is 793.

The employed data is split into 630 train pairs and 163 test pairs of cut-out crops and the assigned identities. To ensure that the testing set does not introduce any new identities, we include all possible identities in the training set. For a fair evaluation on all identities, the train/test split is stratified, i.e., each identity has the same fixed percentage of its cut-out crops allocated to the test set. Consequently, identities with a higher total number of crops will contribute more to the test set compared to identities with fewer crops, ensuring proportional representation across all identities. The corresponding subset on Hugging Face is "chicken-re-id-best-visibility".

To avoid data leakage, it is important to apply data augmentation only after a train-test split is established. This ensures that augmented versions of the same original image do not appear in both sets. We dynamically apply the following data augmentation during training on the "chicken-re-id-best-visibility" subset: rotation, flip, RandAugment [86], and random color-jitter. No data augmentation is applied to the test set.

## 4.2 Baseline approaches

To establish a baseline for the closed set re-identification task, we test three different approaches on our dataset. Each approach involves two steps. First, a feature extractor generates embeddings for the cut-out crops. Second, the resulting feature vectors (embeddings) are then passed to a classifier to ultimately assign the identities. We test each approach with a variation of two classifiers: k-Nearest Neighbor (k-NN) and a linear classifier adapted from the Lightly library [87] (MIT License). All feature extractors were fed with images at an input resolution of 384 x 384 pixels and each approach was run three times. The baseline results were obtained on 64GB shared memory Apple M3 Max Chips (2023) running PyTorch 2.3.0 with MPS acceleration.

**MegaDescriptor**  The employed MegaDescriptor-L-384 [35] (CC BY-NC 4.0 license [88]) is a state-of-the-art feature extractor for animal re-identification from the WildlifeDatasets toolkit (MIT license). It is based on the Swin Transformer architecture [89] and was pretrained on diverse datasets featuring various animal species. However, it has not been trained on chicken data and we did not fine-tune it either. A notable hyperparameter choice made by the MegaDescriptor-L384 authors is the ArcFace [90] loss function, which aims to aid in building meaningful embeddings. We selected the frozen MegaDescriptor-L-384 model over DINOv2 [91] and CLIP [92] due to its better performance on unseen animal domains, as reported by the authors. Their evaluation included cattle as an example of an unseen domain [35].

**Swin Transformer**  We utilize the swin_large_patch4_window12_384 architecture [89] as implemented in [93]. We train it from scratch on the Chicks4FreeID dataset in a fully supervised manner. The training process and hyperparameters mirror those used to build the MegaDescriptor-L384, which also employs the same Swin Transformer architecture. Unlike the frozen MegaDescriptor-L384, which was trained on a variety of animal datasets, we now train the Swin architecture exclusively on our own dataset. The Swin Transformer itself is based on the Vision Transformer architecture.

**Vision Transformer**  Finally, we employ the ViT-B/16 [94] architecture, as implemented in [95], and train it on the Chicks4FreeID dataset in a fully supervised manner with a simple cross-entropy loss. We adopted the effective hyperparameter settings as used in Lightly's benchmarks [87], including optimizer and scheduler choices, for our experiments. The difference between the Swin Transformer

228 and the Vision Transformer lies in how they handle image data; the Swin Transformer uses a
229 hierarchical structure with shifted windows to capture local and global features, while the Vision
230 Transformer treats images as sequences of patches, relying on self-attention mechanisms throughout.

### 4.3 Evaluation

232 For all baselines, we provide three of the most common metrics for closed set animal re-identification.
233 These are: mAP (mean Average Precision), Top-1 accuracy (ratio of correct predictions versus total
234 predictions), and Top-5 accuracy (accuracy of the correct class being within the top 5 predictions) as
235 implemented in TorchMetrics [96].

### 4.4 Baseline results and discussion

237 The results for all baseline approaches and the respective variations are summarized in Table 2.
238 Overall, the experiments yield good results but still leave room for improvement.

Table 2: Baseline results for the closed set re-identification experiments. The highest scores for each metric are in blue.

| Feature extractor | Training | Epochs | Classifier | mAP | Top-1 | Top-5 |
|---|---|---|---|---|---|---|
| MegaDescriptor [35] | pretrained, frozen | - | k-NN | $0.649 \pm 0.044$ | $0.709 \pm 0.026$ | $0.924 \pm 0.027$ |
| MegaDescriptor [35] | pretrained, frozen | - | linear | $0.935 \pm 0.005$ | $0.883 \pm 0.009$ | $0.985 \pm 0.003$ |
| Swin Transformer [89] | from scratch | 200 | k-NN | $0.837 \pm 0.062$ | $0.881 \pm 0.041$ | $0.983 \pm 0.010$ |
| Swin Transformer [89] | from scratch | 200 | linear | $0.963 \pm 0.022$ | $0.922 \pm 0.042$ | $0.987 \pm 0.012$ |
| Vision Transformer [94] | from scratch | 200 | k-NN | $0.893 \pm 0.010$ | $0.923 \pm 0.005$ | $0.985 \pm 0.019$ |
| Vision Transformer [94] | from scratch | 200 | linear | $0.976 \pm 0.007$ | $0.928 \pm 0.002$ | $0.990 \pm 0.012$ |

239 Both the Swin Transformer and Vision Transformer architectures, when trained from scratch, outper-
240 formed the frozen MegaDescriptor model. Additionally, linear classifiers consistently outperformed
241 k-NN classifiers. This indicates that performance scales with the level of supervision, which aligns
242 with expectations.

243 The gap between the MegaDescriptor, a model from a different domain (trained on different species),
244 and those trained from scratch on the target species suggests that the Chicks4FreeID dataset likely
245 has unique characteristics not present in the datasets used to pretrain the MegaDescriptor. Thus, our
246 dataset could enhance the underlying data distribution used to train general animal re-identification
247 models like the MegaDescriptor.

248 Additionally, there is a small improvement in scores between the Vision Transformer over the Swin
249 architecture, which was used to train the MegaDescriptor. The slightly better performance of the
250 Vision Transformer might be due to two reasons: First, we observed a more stable training process for
251 the Vision Transformer (cross-entropy loss) than for the Swin Transformer (ArcFace loss). Therefore
252 we believe that training a more straightforward approach allows for easier convergence on a small
253 dataset like ours. Second, we replaced the standard classification head of the Vision Transformer
254 with a simple linear layer. Since a simple linear layer has limited discriminative power, achieving
255 good overall performance suggests the presence of good embeddings, which was confirmed by the
256 embedding evaluation using k-NN.

## 5 Conclusion

### 5.1 Findings

259 The Chicks4FreeID benchmark dataset was introduced. To the best of our knowledge, it is the very
260 first publicly available dataset for chicken re-identification. The dataset is well-annotated and released
261 under the relatively unrestrictive CC BY 4.0 license. It contains 1270 instance annotations of 54
262 individuals - 50 individuals and 1215 of the instances are chicken. The 677 images, which depict
263 mainly chickens from 11 different coops and various breeds, were individually captured rather than

derived from video. The dataset was created systematically, with manual annotation and instance-to-individual assignments based on expert knowledge, without the use of automated methods, ensuring reliable ground truth annotations. Instead of providing merely bounding boxes that might include parts of the background or other individuals, we offer preprocessed cut-out crops based on precise segmentations of the instances. While the main use case of the dataset is the re-identification of chickens, it also supports semantic and instance segmentation. In addition to instance and semantic segmentation masks, information on identity, animal category, and coop, the dataset also includes a visibility rating of the instances, accounting for occlusions. For the task of closed set chicken re-identification, we established a baseline on the dataset, achieving Top-1 accuracy scores up to 0.928, Top-5 accuracy scores up to 0.990, and mAP scores up to 0.976 with the Vision Transformer. The experiments suggest that the introduced dataset could be a valuable resource for training more robust (general) animal re-identification systems.

## 5.2 Limitations

One clear limitation of the dataset is its size. With 1215 instance annotations of 50 chicken individuals, it is comparatively small. There also exists an imbalance within the classes (individuals), with the number of instances ranging from 4 to 27 in the "best" visibility subset. For chicken breeds with minimal inter-individual variability (e.g., uniform plumage), having more individuals and more instances of each individual would likely aid in re-identification. Additionally, all images of a given chicken were taken on the same day, so changes in appearance over time were not captured. An open question is the dataset's applicability to industrial farming, where thousands of chickens of a single breed are typically kept. A specialized dataset for such breeds could potentially be more suitable for commercial applications. Furthermore, the chicken breeds included in the Chicks4FreeID dataset are not exhaustive, despite their variability. The specific breeds were not annotated because they could not always be accurately determined.

## 5.3 Future work

To further enhance the Chicks4FreeID dataset and address its current limitations, future work could focus on several promising directions. Expanding the dataset to include a larger number of individuals and an even broader range of breeds would enhance its robustness and generalizability. Enriching the metadata with detailed breed-specific information could provide additional context. Methods to automatically create new labeled samples from existing data using generative AI, as proposed in [97], could be evaluated for their potential to aid in expanding the dataset. To capture changes in appearance over time due to factors such as molting, growth, and environmental conditions, individuals from the dataset may be photographed again, provided they are still alive. Similarly, new individuals added to the dataset could be photographed repeatedly over time. The versioning system of the dataset facilitates potential expansions and continuous improvements, ensuring its ongoing relevance and applicability for future research. However, the challenge of long-term data collection persists, as free-range chickens often fall prey to wild predators (e.g., foxes or raccoons). Another interesting direction for future work would be the investigation of models trained on the dataset and their applicability to industrial farming settings with crowded conditions and chickens of a single breed. On a final note, we envision the Chicks4FreeID dataset being utilized by established and aspiring researchers alike, i.e., in future research, contributing to the development of chicken-specific and multi-species re-identification systems, as well as being used for practicing purposes.

## Acknowledgments and Disclosure of Funding

We are immensely thankful to the kind chicken owners who opened their coops for our research, allowing us to collect data and generously offering us fresh eggs. Each of your chickens has made a unique and valuable contribution to the advancement of science.

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
