# Supplementary Material for "Chicks4FreeID: A Benchmark Dataset for Chicken Re-Identification"

**Daria Kern**[1,2] *     **Tobias Schiele**[1,2] *     **Ulrich Klauck**[1,3]     **Winfred Ingabire**[2]

[1]Aalen University, Germany
`{daria.kern, tobias.schiele, ulrich.klauck}@hs-aalen.de`
[2]Glasgow Caledonian University, United Kingdom
`winfred.ingabire@gcu.ac.uk`
[3]University of the Western Cape, South Africa

## 1 Datasheets for datasets

### 1.1 Motivation

**For what purpose was the dataset created?**

The Chicks4FreeID dataset was created specifically for the task of chicken re-identification - i.e., recognizing the identity of an individual chicken in an image. There were two primary motivations for developing this dataset. First, there is a significant need for publicly available and well-annotated datasets in the field of animal re-identification. Second, there was a notable gap, as no such dataset existed for chickens prior to this effort.

However, the dataset is multipurpose and can also be used for semantic segmentation, instance segmentation, or even anomaly detection. It was structured, annotated, and prepared to support these additional tasks effectively.

**Who created the dataset (e.g., which team, research group) and on behalf of which entity (e.g., company, institution, organization)?**

Daria Kern and Tobias Schiele created the dataset.

**Who funded the creation of the dataset?**

The creation of the dataset was not funded by any external sources; it was driven solely by the motivation to create the first of its kind.

**Any other comments?**

No.

---

*contributed equally. Contact: Chicks4FreeID@dariakern.com

Submitted to the 38th Conference on Neural Information Processing Systems (NeurIPS 2024) Track on Datasets and Benchmarks. Do not distribute.

## 1.2 Composition

**What do the instances that comprise the dataset represent (e.g., documents, photos, people, countries)?**

The Chicks4FreeID dataset contains top-down view images of individually segmented and annotated chickens (with roosters and ducks also possibly present). The following tree illustrates the basic structure of the dataset as contained in the "v1_240507.zip" file. However, for a detailed folder structure, see Section 2.4 "Reading the dataset" of the supplementary material.

```
Chicks4FreeID/
└── v1_240507.zip/
    ├── reID/
    │   ├── chicken/
    │   ├── rooster/
    │   └── duck/
    ├── masks
    └── images
```

The main directory can contain different ".zip" files representing different versions of the dataset. Currently, there is only one version available: "v1_240507.zip". However, more versions may be added in the future. The directory corresponding to the version number contains the actual dataset, which is organized into three subfolders: "reID", "masks", and "images".

The "images" folder contains 677 ".png" images, each depicting at least one chicken. Each image has a corresponding color-coded semantic segmentation mask stored in the "masks" folder. Table 1 shows the color codes for the four possible object types. Figure 1 displays an example of such an image and semantic segmentation mask pair.

Table 1: Color codes for each object type.

|  | **Chicken** | **Rooster** | **Duck** | **Background** |
|---|---|---|---|---|
| HEX | #1E1CFF ■ | #FF0000 ■ | #FF4A46 ■ | #FF34FF ■ |
| RGB | (30, 28, 255) | (255, 0, 0) | (255, 74, 70) | (255, 52, 255) |

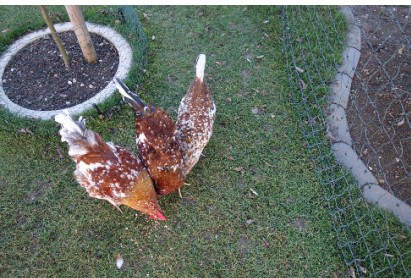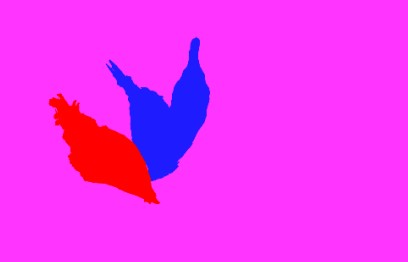

Figure 1: Image (left) with color-coded semantic segmentation mask (right).

Furthermore, the "masks" folder contains binary segmentation mask(s) for the animal instance(s) in the pictures. Figure 2 depicts an example of an image containing three instances and their corresponding instance masks. These instance masks aid the task of instance segmentation and facilitate the preprocessing steps for subsequent animal re-identification.

The "reID" folder contains three subfolders "chicken", "rooster", "duck", each representing a different animal category. These subdirectories hold cut-out and cropped images of the respective animal instances. The cut-out crops result from the preprocessing steps detailed in Section 3.5 "Preprocessing" in the paper. Figure 3 shows an example image alongside its corresponding preprocessed cut-out

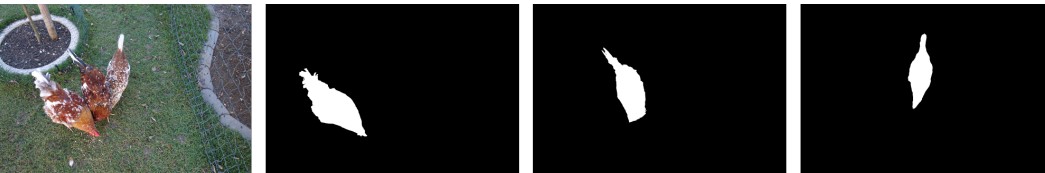

Figure 2: Image (left) with binary segmentation masks (one for each instance).

crops. Note that the crops were squared but not resized during preprocessing and therefore may vary in size.

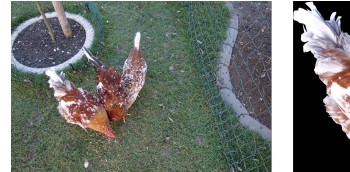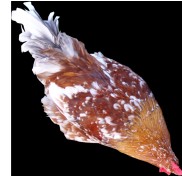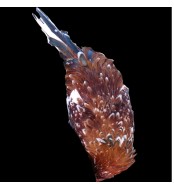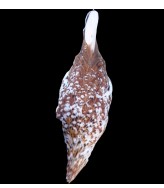

Figure 3: Image (left) with preprocessed cut-out crops (one for each instance).

**How many instances are there in total (of each type, if appropriate)?**

The "images" directory contains 677 images. Whereas the "masks" directory contains 677 semantic segmentation masks and 1270 instance segmentation masks. Table 2 illustrates the number of instances (cut-out crops) in the "reID" directory, sorted by animal category.

Table 2: Number of instances in the "reID" directory.

| Chicken | Rooster | Duck | Total |
|---------|---------|------|-------|
| 1215    | 15      | 40   | 1270  |

**Does the dataset contain all possible instances or is it a sample (not necessarily random) of instances from a larger set? If the dataset is a sample, then what is the larger set?**

The Chicks4FreeID dataset was created entirely anew and is not derived from any existing larger dataset. It features mainly chickens of various breeds. The setting is non-industrial, featuring backyard chickens from 11 randomly selected private households in southern Germany. It is a sample, not an exhaustive collection, and does not fully represent the world's entire chicken population. However, it captures diverse individuals typical of backyard chicken keeping in southern Germany. Figure 4 shows an excerpt from the dataset.

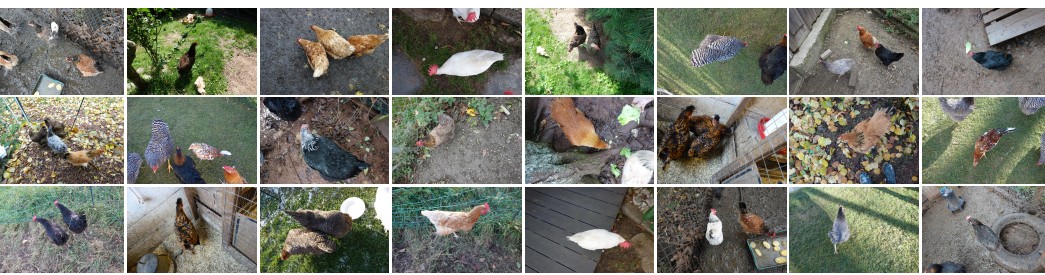

Figure 4: Excerpt from the Chicks4FreeID dataset.

### What data does each instance consist of?

As mentioned above, every animal instance visible in an image is classified into an animal category: "chicken", "rooster", or "duck". The animal instances are further annotated with values assigned for "identity", "coop", and "visibility". The "identity" value denotes the name of the individual, which can be one of 54 predefined names, or "Unknown" if the human annotator could not determine the identity of the animal. The "coop" attribute represents the specific coop to which the animal belongs, with 11 possible numeric values ranging from 1 to 11. Each identity is exclusively associated with a single coop. The "visibility" rating indicates how much of the animal is visible in the segmented instance, with possible values of "best" "good" and "bad" (for an example, see Figure 5). For further information, see Section 3.3 "Annotation" in the paper.

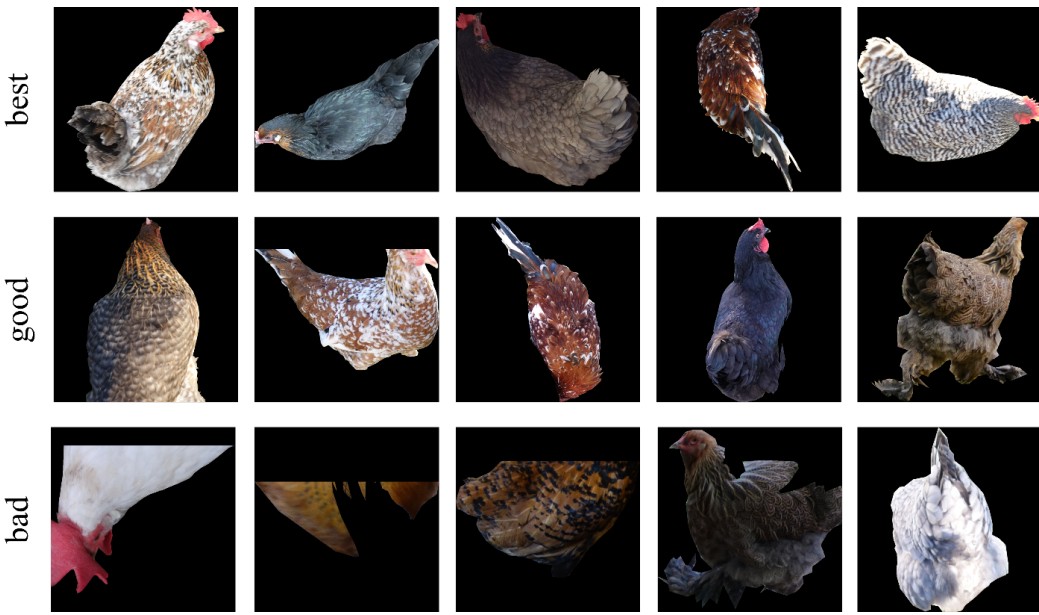

Figure 5: Examples of visibility rating "best", "good" and "bad".

### Is there a label or target associated with each instance?

The target during training varies depending on the specific task at hand (see Table 3). A specific dataset subset configuration was created on Hugging Face for each task. For individual chicken re-identification, use the "identity" value (the assigned name) of segmented instances as the target. However, avoid using the "Unknown" identity as a target. This value does not signify a new and unidentified individual as it would in open set re-identification. Instead, it indicates that the human annotator was unable to assign an identity due to poor visibility. This is also reflected in the fact that an "Unknown" label is only possible in animal instances labeled with a "visibility" value of "bad". Furthermore, exclude all 4 identities belonging to the animal categories "rooster" and "duck". The authors explicitly advise against using roosters and ducks for re-identification tasks. Unlike with chickens, there was no specific focus on roosters or ducks during data collection. As a result, roosters and ducks appear randomly and much less frequently in images. For the same reason, the "rooster" and "duck" animal categories serve as exceptions and could possibly be utilized for anomaly detection tasks. For the task of semantic segmentation, utilize the color-coded masks in the "masks" directory as the target during training. For instance segmentation, employ the binary segmentation masks, which can also be found in the "masks" directory.

Table 3: Intended tasks (as reflected in Hugging Face subset configurations) with targets and inputs.

| Task | Input | Target |
|---|---|---|
| chicken re-identification as in the paper | cut-out crops of "visibility" "best" | 50 chicken "identity" values |
| chicken re-identification (all) | all cut-out crops (excluding "identity" "Unknown") | 50 chicken "identity" values |
| anomaly detection | cut-out crops | animal category "duck" and "rooster" |
| semantic segmentation | images | color-coded segmentation masks |
| instance segmentation | images | binary instance segmentation masks |

**Is any information missing from individual instances?**

The "identity" "Unknown" was assigned to segmentation instances in cases where the human annotator was unable to identify the individual. Unlike in open set re-identification, where this label would suggest a new and previously unseen individual, here it merely indicates that poor visibility prevented the correct annotation. All visible individuals in the Chicks4FreeID dataset belong to a closed set.

**Are relationships between individual instances made explicit (e.g., users' movie ratings, social network links)?**

N/A.

**Are there recommended data splits (e.g., training, development/validation, testing)?**

For the baseline in the paper, we used the "chicken-re-id-best-visibility" subset on Hugging Face. It is divided into 630 training pairs and 163 testing pairs of cut-out crops with assigned identities. All identities have to be included in the training set for the closed set re-identification. To ensure fair evaluation, the train/test split is stratified, meaning each identity has the same fixed percentage of its cut-out crops allocated to the test set. As a result, identities with more crops will contribute more images to the test set than those with fewer crops, ensuring proportional representation across all identities.

**Are there any errors, sources of noise, or redundancies in the dataset?**

To the best of the authors' knowledge, there are none. Should any issues become known, they will be communicated to the dataset consumers accordingly.

**Is the dataset self-contained, or does it link to or otherwise rely on external resources (e.g., websites, tweets, other datasets)?**

It is self-contained.

**Does the dataset contain data that might be considered confidential (e.g., data that is protected by legal privilege or by doctor–patient confidentiality, data that includes the content of individuals' non-public communications)?**

No.

**Does the dataset contain data that, if viewed directly, might be offensive, insulting, threatening, or might otherwise cause anxiety?**

The authors believe it is highly unlikely that the images would be offensive, as they do not originate from commercial farming settings. Caution is advised for anyone suffering from alektorophobia.

**Any other comments?**

For a detailed data composition, see Table 4 and Table 5.

Table 4: Full overview of all chicken annotations in the Chicks4FreeID dataset.

| Coop | #Images | ID | Bad | Best | Good | Total |
|---|---|---|---|---|---|---|
| 1 | 29 | **Coop Total** | **16** | **28** | **5** | **49** |
| | | #Unknown | 11 | 0 | 0 | 11 |
| | | Chantal | 1 | 5 | 0 | 6 |
| | | Chayenne | 1 | 8 | 1 | 10 |
| | | Jaqueline | 1 | 5 | 1 | 7 |
| | | Mandy | 2 | 10 | 3 | 15 |
| 2 | 36 | **Coop Total** | **14** | **39** | **13** | **66** |
| | | #Unknown | 4 | 0 | 0 | 4 |
| | | Henny | 2 | 12 | 4 | 18 |
| | | Shady | 3 | 14 | 3 | 20 |
| | | Shorty | 5 | 13 | 6 | 24 |
| 3 | 60 | **Coop Total** | **22** | **58** | **16** | **96** |
| | | #Unknown | 5 | 0 | 0 | 5 |
| | | Amalia | 3 | 6 | 3 | 12 |
| | | Edeltraut | 2 | 10 | 3 | 15 |
| | | Erdmute | 2 | 12 | 6 | 20 |
| | | Oktavia | 4 | 12 | 3 | 19 |
| | | Siglinde | 4 | 10 | 1 | 15 |
| | | Ulrike | 2 | 8 | 0 | 10 |
| 4 | 26 | **Coop Total** | **7** | **29** | **5** | **41** |
| | | Hermine | 4 | 12 | 5 | 21 |
| | | Matilda | 3 | 17 | 0 | 20 |
| 5 | 116 | **Coop Total** | **84** | **141** | **48** | **273** |
| | | #Unknown | 22 | 0 | 0 | 22 |
| | | Erna | 5 | 12 | 4 | 21 |
| | | Heidi | 10 | 20 | 4 | 34 |
| | | Isabella | 8 | 18 | 7 | 33 |
| | | Kathrin | 7 | 20 | 5 | 32 |
| | | Marina | 15 | 24 | 10 | 49 |
| | | Monika | 11 | 16 | 9 | 36 |
| | | Regina | 5 | 15 | 6 | 26 |
| | | Renate | 1 | 16 | 3 | 20 |
| 6 | 46 | **Coop Total** | **16** | **52** | **12** | **80** |
| | | #Unknown | 3 | 0 | 0 | 3 |
| ... | ... | ... | ... | ... | ... | ... |
| ... | ... | Camy | 3 | 7 | 1 | 11 |
| | | Samy | 8 | 20 | 9 | 37 |
| | | Yin | 2 | 15 | 2 | 19 |
| | | Yuriko | 0 | 10 | 0 | 10 |
| 7 | 42 | **Coop Total** | **1** | **42** | **5** | **48** |
| | | Brownie | 1 | 24 | 2 | 27 |
| | | Spiderman | 0 | 18 | 3 | 21 |
| 8 | 47 | **Coop Total** | **2** | **48** | **15** | **65** |
| | | Brunhilde | 1 | 11 | 0 | 12 |
| | | Fernanda | 0 | 15 | 3 | 18 |
| | | Isolde | 1 | 4 | 12 | 17 |
| | | Mechthild | 0 | 18 | 0 | 18 |
| 9 | 68 | **Coop Total** | **14** | **87** | **13** | **114** |
| | | #Unknown | 1 | 0 | 0 | 1 |
| | | Mavi | 2 | 17 | 1 | 20 |
| | | Mirmir | 1 | 27 | 5 | 33 |
| | | Nugget | 8 | 25 | 2 | 35 |
| | | Skimmy | 2 | 18 | 5 | 25 |
| 10 | 140 | **Coop Total** | **57** | **189** | **36** | **282** |
| | | #Unknown | 23 | 0 | 0 | 23 |
| | | Beate | 3 | 22 | 5 | 30 |
| | | Borghild | 7 | 18 | 3 | 28 |
| | | Eleonore | 6 | 16 | 3 | 25 |
| | | Henriette | 3 | 26 | 4 | 33 |
| | | Kristina | 3 | 21 | 5 | 29 |
| | | Margit | 2 | 18 | 3 | 23 |
| | | Millie | 3 | 19 | 4 | 26 |
| | | Mona | 6 | 26 | 6 | 38 |
| | | Sigrun | 1 | 23 | 3 | 27 |
| 11 | 67 | **Coop Total** | **8** | **80** | **13** | **101** |
| | | Gretel | 5 | 22 | 4 | 31 |
| | | Lena | 1 | 19 | 0 | 20 |
| | | Tina | 2 | 25 | 7 | 34 |
| | | Yolkoono | 0 | 14 | 2 | 16 |
| **Grand Total** | **677** | **50** | **241** | **793** | **181** | **1215** |

Table 5: Full overview of all rooster and duck annotations in the Chicks4FreeID dataset.

| Coop | ID | Category | Bad | Best | Good | Total |
|---|---|---|---|---|---|---|
| 4 | **Coop Total** | | **22** | **3** | **15** | **40** |
| | Evelyn | Duck | 11 | 2 | 9 | 22 |
| | Marley | Duck | 11 | 1 | 6 | 18 |
| 5 | Elvis | Rooster | 6 | 1 | 4 | 11 |
| 9 | Jackson | Rooster | 2 | 1 | 1 | 4 |
| **Grand Total** | **4** | | **30** | **5** | **20** | **55** |

## 1.3 Collection process

### How was the data associated with each instance acquired?

The identities of the subjects were meticulously studied prior to photography, closely monitored throughout the image capture process, and ultimately assigned by a human annotator. No algorithms were used. During photography, the focus was always on a single chicken (the chickens were photographed sequentially, not randomly), while other individuals were able to enter the frame as well.

At first glance, it may appear that chickens of the same breed are indistinguishable (see Figure 6). However, several ways exist to differentiate them visually. For example, examination of the comb reveals differences; chickens may have combs that tilt to the left or right, and the teeth and shapes of these combs also vary (see Figure 7). Additionally, wattle shape and size, patterns in their plumage, body shape, etc. can provide clues to their identities. Figure 8 displays an example of differences in the tail feathers. Fortunately, chickens within the same coop were relatively easy to distinguish (by the human annotator) in most cases. However, there were also cases where identities could not be definitively determined, such as when the comb and significant portions of the plumage were not visible. These instances were labeled as "Unknown".

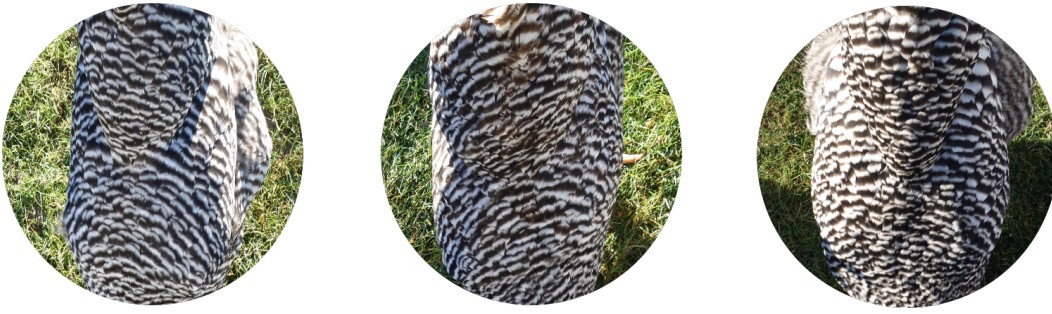

Figure 6: Comparison of chickens of the same breed: individuals Isabella (left), Kathrin (middle), and Marina (right). Minor differences in plumage provide clues to the identity of the chickens.

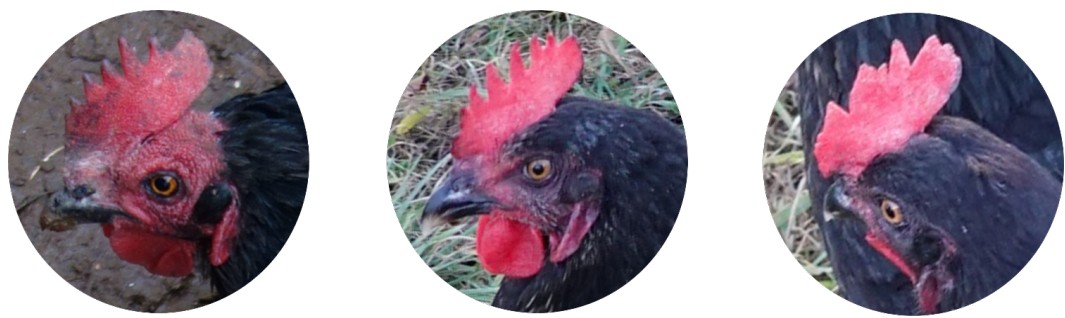

Figure 7: Comparison of different combs: individuals Erdmute (left), Isolde (middle), and Fernanda (right).

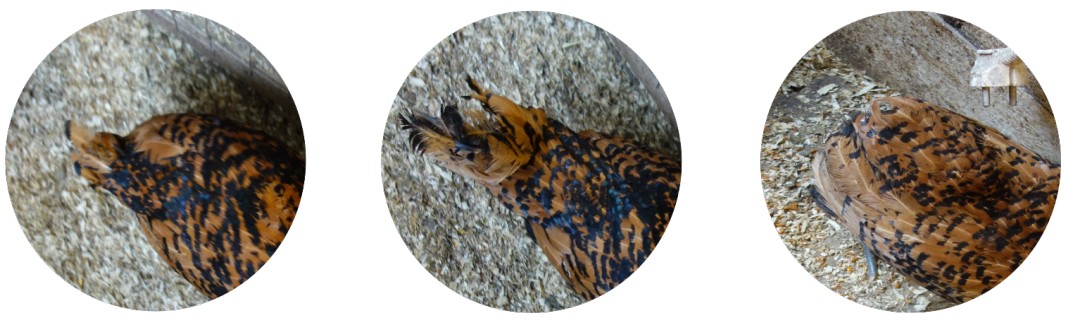

Figure 8: Comparison of different tail feathers: individuals Camy (left), Samy (middle), and Yin (right).

**What mechanisms or procedures were used to collect the data (e.g., hardware apparatuses or sensors, manual human curation, software programs, software APIs)?**

Data was collected manually using two models of cameras: the "Sony CyberShot DSC-RX100 VI" and the "Sony CyberShot DSC-RX100 I".

**If the dataset is a sample from a larger set, what was the sampling strategy (e.g., deterministic, probabilistic with specific sampling probabilities)?**

N/A.

**Who was involved in the data collection process (e.g., students, crowdworkers, contractors) and how were they compensated (e.g., how much were crowdworkers paid)?**

Daria Kern collected the data voluntarily as part of her PhD research, receiving fresh eggs as compensation for her efforts.

**Over what timeframe was the data collected?**

The data collection took approximately one year. However, all images of a coop were always taken within a single day. In other words, all photos of an individual were taken on the same day. Regrettably, backyard chickens frequently fall prey to wild animals such as foxes, raccoons, and predatory birds. This makes it challenging to photograph the same individuals consistently over an extended period.

**Were any ethical review processes conducted (e.g., by an institutional review board)?**

The data collection process was non-intrusive, no animals were harmed, constrained, or put under distress. The owners of the chickens were fully informed about the purpose of the photography and gave their consent before any pictures were taken. They also agreed to the publication of the resulting dataset.

### 1.4 Preprocessing/cleaning/labeling

**Was any preprocessing/cleaning/labeling of the data done (e.g., discretization or bucketing, tokenization, part-of-speech tagging, SIFT feature extraction, removal of instances, processing of missing values)?**

All data were manually labeled by a human annotator (Daria Kern) without any AI assistance. For more information on data annotation, read Section 1.2 "Composition" of the supplementary material and Section 3.3 "Annotation" of the paper. Additionally, data file names reflect the associated labels (see Table 6).

Table 6: File naming additionally reflects the labels.

| Type | File naming + example |
|---|---|
| images | image_\<n\> |
| | image_0 |
| color-coded segmentation masks | image_\<n\>_segmentationMask |
| | image_0_segmentationMask |
| binary instance segmentation mask(s) | image_\<n\>_instanceMask_\<instance\>_coop_\<coop\>_identity_\<identity\>_visibility_\<visibility\> |
| | image_0_instanceMask_0_coop_1_identity_Chantal_visibility_best |
| cut-out crops | image_\<n\>_crop_\<crop\>_coop_\<coop\>_identity_\<identity\>_visibility_\<visibility\> |
| | image_0_crop_0_coop_1_identity_Chantal_visibility_best |

For information on preprocessing, read Section 3.5 "Preprocessing" in the paper.

**Was the "raw" data saved in addition to the preprocessed/cleaned/labeled data (e.g., to support unanticipated future uses)?**

The original images are present in the dataset. They are located in the "images" directory.

**Is the software that was used to preprocess/clean/label the data available?**

The software "Labelbox" (available at https://labelbox.com/) was utilized under a free educational license for manual data annotation. No AI-based labeling support was used.

Preprocessing took place before uploading the dataset to Hugging Face. The resulting cut-out crops are part of the dataset and were generated directly from the raw images (which are also part of the dataset) and the Labelbox-annotations. The code is documented on GitHub. For privacy reasons, the API key for accessing the Labelbox-annotations is not included.

**Any other comments?**

No.

## 1.5 Uses

**Has the dataset been used for any tasks already?**

The dataset has been used for closed set re-identification of 50 chickens as described in Section 4 "Experiments" in the paper.

**Is there a repository that links to any or all papers or systems that use the dataset?**

Papers or systems using the dataset will be listed here https://github.com/DariaKern/Chicks4FreeID.

**What (other) tasks could the dataset be used for?**

Section 1.2 "Composition" in the supplementary material talks about the targets associated with each task (see "Is there a label or target associated with each instance?"). Different Hugging Face subset configurations allow the use of the dataset for different tasks (see Table 7).

Table 7: Dataset configurations for different tasks as provided on Hugging Face.

| Hugging Face subset | Task | Modality | | | | Animal Category | | | Visibility | | | Split |
|---|---|---|---|---|---|---|---|---|---|---|---|---|
| | | images | seg. masks | inst. masks | cut-out crops | chicken | rooster | duck | best | good | bad | |
| chicken-re-id-best-visibility | 1 | | | | X | X | | | X | | | train + test |
| chicken-re-id-all-visibility | 2 | | | | X | X | | | X | X | X | train |
| animal-category-anomalies | 3 | | | | X | X | X | X | X | X | X | train |
| instance-segmentation | 4 | X | | X | | X | X | X | X | X | X | train |
| semantic-segmentation | 5 | X | X | | | X | X | X | X | X | X | train |
| full-dataset | 6 | X | X | X | X | X | X | X | X | X | X | train |

Tasks:

1. closed set re-identification of 50 chicken as described in the paper.
2. super difficult closed set re-identification of 50 chicken (contains instances of bad visibility). However, identitiy "Unknown" is excluded.
3. anomaly detection (anomalies = roosters + ducks).
4. instance segmentation.
5. semantic segmentation (classes = chicken, rooster, duck, background).
6. custom task.

**Is there anything about the composition of the dataset or the way it was collected and preprocessed/cleaned/labeled that might impact future uses?**

N/A.

**Are there tasks for which the dataset should not be used?**

The dataset should not be used for duck or rooster re-identification.

**Any other comments?**

No.

## 1.6 Distribution

**Will the dataset be distributed to third parties outside of the entity (e.g., company, institution, organization) on behalf of which the dataset was created?**

Yes, it is publicly available on the internet.

**How will the dataset will be distributed (e.g., tarball on website, API, GitHub)?**

The Chicks4FreeID dataset can be accessed here:

- Dataset: https://huggingface.co/datasets/dariakern/Chicks4FreeID
- DOI: https://doi.org/10.57967/hf/2345

**When will the dataset be distributed?**

The Chicks4FreeID dataset was first released in 2024.

**Will the dataset be distributed under a copyright or other intellectual property (IP) license, and/or under applicable terms of use (ToU)?**

The Chicks4FreeID dataset is distributed under the CC BY 4.0 license.

**Have any third parties imposed IP-based or other restrictions on the data associated with the instances?**

No.

**Do any export controls or other regulatory restrictions apply to the dataset or to individual instances?**

No.

**Any other comments?**

No.

## 1.7 Maintenance

**Who will be supporting/hosting/maintaining the dataset?**

Daria Kern and Tobias Schiele will support and maintain the dataset. The dataset is hosted on Hugging Face and has its own DOI ( https://doi.org/10.57967/hf/2345).

**How can the owner/curator/manager of the dataset be contacted (e.g., email address)?**

The curators of the data set can be contacted by email Chicks4FreeID@dariakern.com.

**Is there an erratum?**

Not to our knowledge.

**Will the dataset be updated (e.g., to correct labeling errors, add new instances, delete instances)?**

Any new versions will be uploaded to Hugging Face into the same repository but under a different version number. Updates will be communicated on the GitHub and Hugging Face repositories.

**If the dataset relates to people, are there applicable limits on the retention of the data associated with the instances (e.g., were the individuals in question told that their data would be retained for a fixed period of time and then deleted)?**

While each of the chickens has their own unique personality, they are not considered people.

Yes. Versioning of the dataset is supported, and future versions will be marked as such, while older versions will be maintained.

No. However, this may change in the future.

No.

## 2 Dataset and Code

### 2.1 Access

- Dataset: https://huggingface.co/datasets/dariakern/Chicks4FreeID
- DOI: https://doi.org/10.57967/hf/2345
- Croissant metadata: https://huggingface.co/api/datasets/dariakern/Chicks4FreeID/croissant
- Code: https://github.com/DariaKern/Chicks4FreeID

### 2.2 License

The Chicks4FreeID dataset and the accompanying code (excluding imported libraries or models from external sources, which have their own licenses) are released under the CC BY 4.0 license. This license allows for the distribution, remixing, adaptation, and building upon the dataset in any medium or format. Users must give appropriate credit to the authors, include a link to the license, and clearly indicate if any changes were made. Commercial use of the dataset is permitted. For more information, please visit https://creativecommons.org/licenses/by/4.0/.

Statement of responsibility: The authors declare that they bear all responsibility for violations of rights. They also confirm that this dataset is released under the CC BY 4.0 license.

### 2.3 Quick dataset overview

Modalities:

- 677 images
- 677 color-coded semantic segmentation masks
  (classes: chicken, rooster, duck, background)
- 1270 binary instance segmentation masks
- 1270 preprocessed cut-out crops

Annotations:

- Animal category (chicken, rooster, duck)
- Identity (54 unique names)

- Coop (1-11)
- Visibility (best, good, bad)

Uses:

- chicken re-identification
- instance segmentation
- semantic segmentation
- (anomaly detection)

## 2.4 Reading the dataset

**Dataset .zip file** The "v1_240507.zip" file can be downloaded on Hugging Face. It contains the whole Chicks4FreeID dataset. The original images are in the "images" folder. Instance and segmentation masks can be found in the "masks" folder. The reID folder, containing the preprocessed cut-out crops, is arranged as follows: First, the folders are divided into the three animal categories (chicken, rooster, duck).

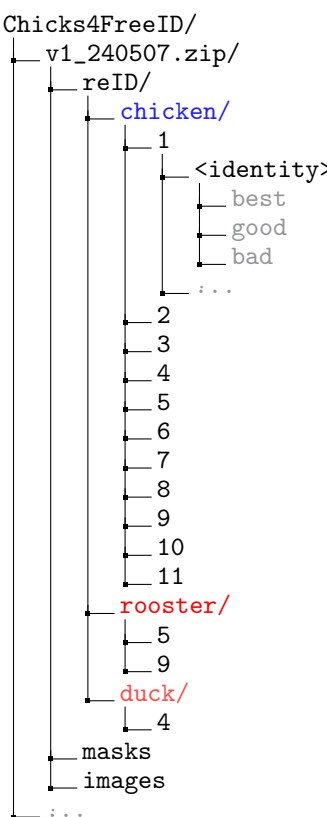

Since chickens are present in every coop, the "chicken" folder includes a separate subfolder for each of the 11 coops. Roosters and ducks, being absent in most coops, have fewer subfolders as a result.

The numbered coop folders, in turn, contain subfolders named after the individuals living in them. Some, but not all, also contain a subfolder named "Unknown", indicating instances with unassigned identities. For information about "Unknown", read Section 1.2 "Composition" question "Is any information missing from individual instances?".

The identity folders are further divided into final subfolders that contain the cut-out crops. The visibility level of the instances (visible on the cut-out crops) is indicated by the name of the folder

in which they are in. If cut-out crops of a certain visibility level do not exist for an individual, the corresponding folder will not be present.

**The Hugging Face** `pip install datasets` **library**    Another preferable option is to directly access the dataset with the Hugging Face library. This library manages caching, and loading and allows accessing splits and subsets of the dataset. To install the required package, use the following command in your terminal:

```
pip install datasets
```

To load the data, use the following Python code:

```python
from datasets import load_dataset
train_ds = load_dataset("dariakern/Chicks4FreeID", split="train")
train_ds[0]
```

The output of the above code will be:

```
{'crop': <PIL.PngImagePlugin.PngImageFile image
        mode=RGB size=2630x2630 at 0x7AA95E7D1720>,
 'identity': 43}
```

The above code loads the train split of the default subset configuration, which is named `chicken-re-id-best-visibility`. See Table 7 in Section 1.5 "Uses" of the supplementary material for the modalities of each subset configuration. To load the test split or to load other subsets, type:

```python
repo = "dariakern/Chicks4FreeID"
ds = load_dataset(repo, split="test")   # Change split
ds = load_dataset(repo, "chicken-re-id-all-visibility")
ds = load_dataset(repo, "chicken-category-anomalies")
ds = load_dataset(repo, "instance-segmentation")
ds = load_dataset(repo, "semantic-segmentation")
ds = load_dataset(repo, "full-dataset")
```

For more information on how to work with `datasets`, please visit the official documentation for Hugging Face datasets.

**Croissant**    Hugging Face also provides a mlcommons/croissant metadata export.    For that, click the croissant tag on the Hugging Face page of the Chicks4FreeID dataset: https://huggingface.co/api/datasets/dariakern/Chicks4FreeID/croissant.

## 2.5    Reproducing the baseline

**Requirements and licenses**    Below, the requirements for replicating the baseline results are shown with their respective versions and licenses.

```python
# For loading the Chicks4FreeID dataset
datasets==2.19.1   # Apache2.0
# For benchmarking utils
lightly==1.5.2   # MIT
# For logging and calculating metrics
```

```
matplotlib==3.8.4   # BSD Compatible
tensorboard==2.16.2   # Apache2.0
pandas==2.2.2   # BSD-3
torchmetrics   # Apache2.0
# For model building / loading / training
timm==0.9.16   # MIT
torch==2.3.0   # BSD-3
# For the ArcFace loss
wildlife-tools==0.0.2   # MIT
# Second level dependencies (not automatically installed)
tabulate==0.9.0   # of pandas (GPL-2.0)
pytorch-metric-learning==2.5.0 # of wildlife-tools (MIT)
psutil   # (BSD-3)
```

**Baseline**    To clone the repository and run the baseline script, use the following commands in your terminal:

```
git clone https://github.com/DariaKern/Chicks4FreeID
cd Chicks4FreeID
pip install requirements.txt
python run_baseline.py
```

You can pass different options to the script, depending on your hardware configuration:

```
python run_baseline.py --devices=4 --batch-size-per-device=128
```

For a full list of arguments, type:

```
python run_baseline.py --help
```

In a separate shell, open TensorBoard to view the experiments' progress and results:

```
tensorboard --logdir baseline_logs
```

**Note**    Different low-level accelerator implementations (TPU, MPS, CUDA) yield different results. The original hardware configuration for the reported results is based on the MPS implementation accessible on a 64GB Apple M3 Max chip (2023). It is recommended to execute the baseline script with at least 64GB of VRAM / Shared RAM. Using the described device, one run takes around 9:30h.

**Supplementary details**    This paragraph provides supplementary details not found in the paper about the usage of `torchvision.transforms`. Table 8 shows the detailed transforms applied in each data loader. Note that the table shows the train sets, on the testing set, none of these data augmentations have been applied; only the respective normalization transform is used in each case. The shortcuts stand for:

- ROT: Random Rotation (360 degrees)
- FLIP: Random Horizontal and Vertical Flip
- CJ: Color Jitter

- RA: RandAugment (`torch.transform.RandAugment`)
- IMG: ImageNet Normalization
- NORM: Standard normalization ($mean = 0.5$, $std = 0.5$)

Table 8: Detailed data augmentation and transforms applied on the training split for fitting the models and their corresponding embedding evaluations.

| MegaDescriptor-L384 | | ViT-B/16 | | Swin-L-384 | |
|---|---|---|---|---|---|
| NO TRAINING | | ROT | | ROT | |
| | | FLIP | | FLIP | |
| | | CJ | | RA | |
| | | IMG | | IMG | |
| k-NN | Linear | k-NN | Linear | k-NN | Linear |
| ROT | ROT | ROT | ROT | ROT | ROT |
| FLIP | FLIP | FLIP | FLIP | FLIP | FLIP |
| NORM | NORM | IMG | IMG | IMG | IMG |

In other words, we added random rotation and flipping to all training cases. The rationale is that the model should learn invariance to rotation and flips as the chickens are photographed from a top-down view.