# OpenReview forum: "Chicks4FreeID: A Benchmark Dataset for Chicken Re-Identification"
_NeurIPS.cc/2024/Datasets_and_Benchmarks_Track — Submitted to NeurIPS 2024 Track Datasets and Benchmarks_

### Official Review · Reviewer_AJWF · 2024-07-22
**The paper presents a unique and interesting dataset about chickens, well motivated and well evaluated but missing some key details regarding choice of data collection and results to make paper stand out.**

**Rating:** 6
**Confidence:** 3
**Correctness:** The submission looks correct and cons…

**Review:**

The motivation of the paper is clear and the data collection process is also well documented.
One of the first contributions is a review of the state of the art in animal reidentification. The paper includes details about the problem itself and the discusses the datasets available to tackle the problem.

The details about the dataset collection are well documented, the dataset contains top-down images of one or more instances of chickens. The masks are provided for semantic segmentation and instance segmentation. The data is divided into chicken, rooster, and duck categories. The authors have not clearly emphasized the importance of having the two ducks and roosters in the dataset. There is a statement that it might make the data challenging however I would question the decision of adding these extra four animals unless authors can show that having them in the dataset is better than not having them.

The composition of the data is well demonstrated to a point. The images in the dataset are divided based on visibility and some images have clear differences in terms of visibility of the animal in the image. An important note is that images collected are from a single day and top-down view, thus it is not sure how much the data differs in terms of enhancing the complexity in terms of appearance from separate days or separate angles. Typically, reidentification is required to have good accuracy over multiple days and different conditions. This aspect is not captured in the dataset and thus questions the effectiveness of the dataset in terms of making a solid contribution towards problems of re-identification. It is not doubt that such datasets do not exist with chicken and we can consider this as a first step. However, the rationale behind making the dataset in this particular way is not very clear. The authors themselves collected data and therefore lack of variation in terms of appearance should be explained or proven that sufficient variation does exist. I would prefer that authors to explain this particular choice in the paper (i.e. single-day data collection, consistent angle data collection).

The single angle also brings the issue of how to avoid similar-looking images of the same animal in train and test sets. This is not particularly addressed.

Evaluation of the task is done using three state-of-the-art methods. My worry is that dataset seems to be too perfect or too simple for vision transformers. The other networks also show comparable results (at least mAP) with linear classifiers.  The best result is already at 0.976. This is not discussed in the paper which is a major weak point. The result suggests that the problem can be solved with existing methods and therefore the job of the authors shifts to provide other concrete evidence where the dataset can improve state of the art or at least discuss directions where the dataset will improve SOTA. The result is definitely encouraging for people wanting to study chickens but the claim for the dataset to be pushing future research needs to be explained further including limitations mentioned above i.e. similar perspective and lack of multi-day data.

The idea of using mega descriptor with frozen weights is good but it is not clear why authors have not finetuned the model to their data. Arguably learning feature embedding from other datasets should make the re-id stronger. It is worth investigating or at least explaining why it is not performed.

The humor in the acknowledgment is appreciated. I hope you enjoyed the eggs.

The authors have also written very good supplementary material with details of using the dataset and the code.

**Strengths:**

The first dataset in the direction of chicken reidentification.

Large dataset with a relatively high number of individuals. The paper is well structured and the short lit. review of reidentification papers is useful.

The results of the experiments show interesting outcomes when different models are used with pre-training or training from scratch. The results show that dataset-relevant implementation is necessary to get good results. There results are excellent motivation for people working with chickens to use AI.

The supplementary material is well documented.

**Additional Feedback:**

The paper is indeed really good but it is missing some key aspects at the moment. I would suggest to improve them and make the paper stronger. The paper has a good chance of acceptance with more clarifications.

**Clarity:**

The paper is well-written and easy to understand. The authors have maintained a good flow.

Additional clarity is required for the choice of making data in this particular way (addressed above). I have tried to provide some venues for improvement.

**Documentation:**

The documentation effort is done well, supp material is also provided with good details.

**Ethics:**

The authors have taken necessary precautions and mentioned that owners of chickens were informed about the process.

**Limitations:**

The authors have outlined some of the limitations of the work in detail and honestly i.e. images taken on a single day, chickens looking visibly different due to different breeds.

The authors have mentioned that the dataset is also useful for semantic segmentation and instance segmentation. However, besides mentioning the potential use for such problems it would be nice to have a more concrete idea of how and why such datasets add value. One does not have to focus on it too much but since the claim is made some material or text is necessary to support the claim adequately.

**Opportunities For Improvement:**

The authors are not making any claim for the possible use of methods in re-identification for animal behavior experiments. Behavior experiments are many times organized with a top-down view to understand the behavior of animals alone or in a group. Leadership, social structure, hierarchy, mate choice etc. many topics are studies with chickens. The dataset is also collected from a top-down view and therefore anyone working with chickens might benefit from using this dataset for pre-training their algorithms. This should be included in the paper to improve the applicability of the paper.

The authors can improve discussions on the results. The author can discuss the choice of making such a dataset in this specific way. For example, Why use a top-down approach? Why images are taken on a single day, not on different days? if so then did authors attempt to take in different light conditions?

The authors have provided data for re-identification but the discussion on the dataset does not tell the reader how the authors tried to make the dataset challenging. This is necessary because the results are very good, when would this scenario be more challenging? For example, if fewer training images are used during training does the result become worse? Single-shot? OR is it possible that train/test bleeding has happened which has inflated the results due to a lack of diverse perspectives of the same animal?

It would be great if authors could provide the performance accuracy for good - bad - best images. The results might teach something in terms of which set of images has the worst performance. This may also give a better clue on what models are picking up and what is missing, especially between ViT and MegaDescriptor.

The argument against not taking images over multiple days (predation from fox etc.) is valid but over multiple coops on the scale of 50 individuals highly unlikely to have all been dead or hunted within the scale of weeks. It is rather true that data over multiple years can be tricky but one can imagine having data over multiple weeks or months. As a last resort data at different times of day dusk-dawn or some light condition variation will naturally add more complexity.

**Relation To Prior Work:**

Ln 27-28 citations from biology about disease transmission or livestock management with efficient systems leading to improvement in welfare. Authors have cited computer vision paper and it is not sufficient.

**Summary And Contributions:**

The authors offer the first publicly available dataset for solving the problem of chicken reidentification, semantic segmentation, and instance segmentation.
The dataset contains 50 unique individuals with 1215 annotations with 1270 instances in total. The datasets are collected from scratch by authors who have visited private coops of people to picture the animals.
The dataset is motivated by the need for possible automation from the domesticated chicken industry in the context of improving the welfare of the animals in livestock management.
The authors claim that identification applications could also be useful for learning the social interactions of the animals. In general, broadly contributes to improving the re-identification algorithms. The contribution also includes preparing baseline results with common approaches for the problem of re-id.

---

> ### Author Rebuttal · Authors · 2024-08-15
>
> Dear reviewer,
>
> Thank your for your extensive and detailed review and for paying attention to the details in our supplementary material. We are pleased to hear you found the literature review of re-ID papers useful, and the results of our experiments to be an “excellent motivation for people working with chickens to use AI”. Your suggestions were incredibly helpful, and we will certainly incorporate your feedback to clarify any missing aspects, with the aim of making our paper stronger.
>
> # Variation
> **Camera perspective**
>
>
> We aimed for a top-down view as it is more suitable to monitoring settings where the camera is usually placed above the individuals. However, as reviewer 96pm has already correctly mentioned, the top-down view is not clean like a drone shot. The animals were photographed from a standing position (camera did not perfectly point towards the ground), aiming to capture a top-down perspective. However, since the animals were free to move, their distance from the camera varied, which affected the view of their plumage. Ultimately this gives the dataset more variation and makes it more challenging.
>
> **Time span**
>
>
> We acknowledge this limitation and have already addressed it briefly in our paper. While visiting all the coops took approximately one year, all images of a coop were always taken within a single day. This means changes in appearance over time were unfortunately not captured.
>
>
> The practical challenges that influenced our decision to photograph a given chicken on a single day were the following. A number of the chickens had indeed already passed away relatively quickly within weeks, with one even being taken by a fox the night before the scheduled photography session. The uncertainty about the chickens fate led us to take pictures of a given coop on a single day. During the time we collected the data, we received several messages informing us that individual chickens and sometimes entire coops had already died.
>
>
> We really like your suggestion to photograph the chickens from dawn to dusk. This idea is a good compromise, and we now regret not implementing it.
>
> **Visibility rating and occlusion**
>
>
> It was common for animals to be partially obscured by other animals or objects in the images. Therefore all instances are annotated by visibility: "best" (instances with the clearest view of the hens' backs/plumage, at most minimal occlusion), "good" (minor occlusion, neither best or bad category), and "bad" (side views or major occlusion). Occlusion complicates the identification and tracking of individuals. [1] The "best" "good," and "bad" ratings therefore represent different levels of difficulty for re-ID.
>
>
> This annotation allows the user to create their own train-test split and difficulty level from the “full-dataset” subset on Hugging Face. One could, for example, train only on the “good” images and then test on the “bad” images. Various combinations are possible.
>
>
> **Unconstrained movement**
>
>
> The animals were able to move freely at all times, with their poses remaining unconstrained. They exhibited behaviors such as pecking, preening, fluffing their feathers, and looking around. Movement sometimes resulted in blurring of the images of a given chicken. Overexposure due to camera flash, shadows on the plumage or bright light can also be found in instances of the same individual.
>
>
> The attached PDF shows two examples of varying instances of the same individual.
>
> # Experiments
> **Single shot approach**
>
>
> The results do indeed become worse in the single-shot setting (See attached PDF).
>
> **Domain transfer via finetuning**
>
>
> As suggested, we conducted additional promising experiments (See attached PDF).
>
> Fine tuning/domain transfer is particularly effective  (top-1 accuracy 56.52%)when the training dataset is reduced to only one sample per individual. The fine tuned model achieves the best top-1 accuracy (56.52%) in the one shot setting, whereas the Vision Transformer only achieves 50.07% now.
>
>
> **Separate performance accuracy for good / bad / best**
>
>
> We like this idea and would like to include the performance separately for the visibility levels. Due to time constraints, we unfortunately could not achieve this until the rebuttal deadline (16th) . However, we will try to work it into the updated version of the paper.
>
> # Other
> **Semantic and instance segmentation with the Chicks4FreeID dataset**
>
>
> The Chicks4FreeID dataset includes the necessary components (instance masks + semantic segmentation masks) to perform these tasks.
>
> **Roosters and ducks**
>
>
> In the experiments (and the respective subsets on Hugging Face), the “rooster” and “duck” instances are specifically not included. However, they do appear in some images since the animals were allowed to move freely. We did not consider it necessary to discard these images entirely. Instead, we have annotated the visible instances accordingly.
>
> **Behavior experiments**
>
>
> Thank you for this valuable comment about the potential value of our work for behavior experiments. We addressed this in the rebuttal for reviewer mGJR and will also include this in the motivation in the updated version of the paper.
>
> # Rebuttal experiments, results and addressing common concerns of the reviewers
> Please see rebuttal for reviewer cXUF.
>
> # References
> [1] N. Li, Z. Ren, D. Li, L. Zeng, Review: Automated techniques for monitoring the behaviour and welfare of broilers and laying hens: towards the goal of precision livestock farming, Animal, Volume 14, Issue 3, 2020, Pages 617-625, ISSN 1751-7311, https://doi.org/10.1017/S1751731119002155

---

> > ### Comment · Reviewer_AJWF · 2024-09-01
> >
> > Thanks for all responses. I intend to keep the score because of two main concerns initially raised during review but not discussed later in rebuttal. Upon reviewing all answers, I am not super convinced.
> >
> > 1. Results are too good with SOTA, hence no clear idea on where to go from here with this dataset.
> > 2. The diversity in dataset is too low, any method developed on this dataset can not claim or guarantee reidentification between days or even within a day because this diversity does not exist in the given dataset.

---

### Official Review · Reviewer_96pm · 2024-07-24
**A Valuable But Niche Dataset for Chicken Re-Indentification**

**Rating:** 7
**Confidence:** 4
**Clarity:** The paper is concise and well written.

**Review:**

Pros:

- High-quality instance masks of chickens from a top-down view
- First of its kind dataset for a thus far underrepresented animal class (despite intensive use of chickens as livestock across the globe)
- Can contribute to animal welfare through automated monitoring

Cons:

- Fairly small dataset with limited diversity of chicken breeds, potentially limiting generalizability and making it niche
- Different chicken breeds not annotated, unclear how many different breeds are included
- All images of specific individuals were collected on the same day, which does not allow for testing longitudinal re-identification of the same individuals

**Strengths:**

This dataset has several strengths, including high-quality hand-labeled cut-outs instead of mere bounding boxes of chickens. The authors provide a comprehensive review of re-identifications models and datasets for several animal species and demonstrate that chickens (and birds more broadly) are underrepresented, despite their extensive use as livestock around the world. The dataset can contribute to animal welfare by helping with the non-invasive monitoring of chickens in sanctuaries, coops or farms through individual identification. Most of their presented model architectures achieve high performance.

**Additional Feedback:**

The authors state that images are taken from a top-down view. While all images appear to be taken from an elevated position, many of the examples do not look like a clean top-down view (like a drone shot). Could the authors address the variance of angles that the pictures were taken from and discuss implications for the dataset?

It is mentioned that the 11 households with chicken coops were randomly selected. What did this random selection look like?

Can the authors provide a confusion matrix for the different individuals? Is it more likely that individuals with similar plumage coloration are more easily confused?

Similarly, can the authors provide information on which the most relevant features are for re-identification (e.g., body shape/size, plumage coloration, comb shape, ...)?


Supplementary Materials:

Line 53 & 55: change “southern Germany” to “Southern Germany”

**Correctness:**

The dataset is constructed meticulously, and all methods, models and benchmarks are easy to follow and evaluate.

**Documentation:**

The data collection is well described. The authors further provide detailed and clear documentation  and licensing of their dataset, which is available on HuggingFace and Github. The maintenance plan  is laid out clearly.

**Ethics:**

The dataset raises no ethical concerns.

**Limitations:**

The authors addressed the limited size of the dataset (only 677 images), including the unevenly distributed number of instances for different individuals.

The narrow focus of the dataset is a potential limitation. The authors also =addressed the limitations of having all images of specific individuals taken on the same day.

The authors point out the relevance of their dataset for livestock farming. However, there is little discussion about the breeds represented in the dataset compared to the breeds most commonly used as livestock. For example, common breeds for meat (Cornish cross chicken) and egg production (Leghorn chicken) have white plumage. This would presumably make it much more difficult to identify individuals by plumage.

**Opportunities For Improvement:**

The basic terminology to describe chickens vs. hens vs. roosters should be improved. The term chicken encompasses both hens (females) and roosters (males). However, hens are called chickens and roosters are mentioned as a separate animal category, despite belonging to the same species. The authors should clarify their used terminology.

The authors do not provide specific information about the breed of each chicken. They address this point by stating that specific breed information could not be obtained from the owners. However, the supplementary materials nonetheless make statements about how chickens from the same breed can nonetheless be told apart (e.g., Figure 6). Even if the specific breed of each chicken cannot be determined, the authors should at least attempt to give an estimate of how many/which breeds are represented in the dataset and possibly group different individuals by plumage color or other relevant physical features typically associated with breeds.

**Relation To Prior Work:**

The authors discuss re-identification datasets for different animal species and point out that birds are underrepresented. There are no datasets available for chicken re-identification, despite the extensive use of this species. The authors deliver a straightforward argument for how their dataset is filling a relevant gap for animal re-identification. Additional information on the diversity of chicken breeds (and their physical features) used across the world would potentially be helpful to address how generalizable the dataset might be despite its limited size.

**Summary And Contributions:**

The authors introduce Chicks4FreeID, a dataset containing 677 images of chickens (50 hens and two roosters) and two ducks. The dataset consists of meticulously hand-labeled and accurate cut-out crops (N=1270) and binary instance masks (N=1270) of chickens. The paper develops baseline models for re-identification of chickens comparing three different approaches (MegaDescriptor, Swin Transformer, Vision Transformer), with most achieving high accuracy. This dataset can contribute to the monitoring and identification of farm and sanctuary animals.

---

> ### Author Rebuttal · Authors · 2024-08-15
>
> Dear reviewer,
>
> thank you for taking the time to review our work. We are pleased that you found the paper concise, well-written, and the methods easy to follow and evaluate. Especially your idea to group the hens by plumage is very much appreciated. We hope that our responses to your questions also meet your expectations and are satisfactory.
>
> # Breeds
> Current estimates suggest the dataset contains at least 13 different breeds (see the following list). In the updated version of the paper, we will provide more detailed information on the breed aspect. In the meantime, we will continue to gather more information.
>
> + Brahma (confirmed)
> + Appenzeller Spitzhaube (confirmed)
> + Swedish Flower Hen (confirmed)
> + Leghorn (confirmed)
> + Welsummer (unconfirmed)
> + Norfolk Grey (unconfirmed
> + White Sussex (unconfirmed)
> + New Hampshire (unconfirmed)
> + Golden Comet (unconfirmed)
> + Isa Brown (unconfiremed)
> + Australorp (unconfirmed)
> + (Barred) Plymouth Rock (unconfirmed)
> + Legbar (unconfirmed)
>
>
> Thank you for the great idea to group the hens by plumage. We visibly grouped them by color (white, black, orange, gray, brown, stipes, dots, …) in a first attempt in the attached PDF (Unfortunatley the overview is quite small due to the one page limit of the PDF. We apologize for that). We are going to refine this and add the information to the supplementary material in an extra chapter in the updated version.
>
>
> Additionally we are going to add a paragraph where we discuss breeds commonly used as livestock and also address their representation in the dataset.
>
> # Experiments
> As suggested, we included a confusion matrix in the attached PDF.
>
> # Other questions
> **Most relevant features for re-identification**
>
>
> Chickens can perceive a broad spectrum of colors, and their ability to recognize each other is influenced by visible features. The comb and the plumage play a crucial role in individual recognition. Experiments have shown that alterations to the comb can lead to a failure in recognition. This effect is also observed with changes to the plumage. [1]
>
>
> In our dataset we put the focus on the plumage. We have ensured that certain features, such as the feet, are excluded, as they could reveal identity clues due to the presence of leg bands. For the same reason, we also removed the background.
>
>
> We have not tested yet, which features the models prioritize. This could be an interesting direction for future work. We suppose, a hens shape does not play a significant role for the models, as the shape can vary greatly even within instances of the same hen due to its pose and the "not clean" top-down view.
>
> **Tow-down view**
>
>
> As you correctly observed, the top-down view is not clean like a drone shot. The animals were photographed from a standing position, aiming to capture a top-down perspective. However, since the animals were free to move, their distance from the camera varied, which affected the view of their plumage. Ultimately this gives the dataset more variation and makes it more challenging.
>
> **Selection of households**
>
>
> The chicken owners learned about the project through our personal and private network and voluntarily expressed their interest. Appointments were scheduled on a first-come, first-served basis, with no special selection process implemented.
>
> **Terminology**
>
>
> You are absolutely right, hens and roosters are both chicken. We are going to update the terminology in the paper accordingly.
>
> **Typo in line 53 & 55**
>
>
> We changed “southern Germany” to “Southern Germany”
>
> # Rebuttal experiments, results and addressing common concerns of the reviewers
> Please see rebuttal for reviewer cXUF.
>
> # References
> [1] A. M. Guhl and L. L. Ortman, “Visual Patterns in the Recognition of Individuals among Chickens,” The Condor, vol. 55, no. 6, pp. 287–298, 11 1953. [Online]. Available: https://doi.org/10.2307/1365008

---

> > ### Comment · Reviewer_96pm · 2024-09-01
> >
> > Thank you for adding the additional details on breeds, a confusion matrix, and addressing the questions raised in my review. Based on your responses and edits, I raised the score from a 6 to a 7.

---

### Official Review · Reviewer_mGJR · 2024-07-25
**Review of "Chicks4FreeID: A Benchmark Dataset for Chicken Re-Identification"**

**Rating:** 3
**Confidence:** 5
**Correctness:** Yes
**Clarity:** Yes

**Review:**

Pros:
-It introduces the first publicly available dataset dedicated to chicken re-identification, filling a significant gap in the field.
-The authors provide a baseline for closed set re-identification, facilitating comparative analysis for future research.

Cons:
-The necessity of researching chicken re-identification (reid) has not been justified. The lack of publicly available datasets for chicken reid does not imply a pressing need for development in this field; rather, it suggests that this problem may not be worth investigating.
-The inherent value of chickens is limited, and the cost of researching this topic far outweighs its intrinsic value. If the reid were focused on endangered species, it might be more meaningful.
-As shown in Table 2, the reid success rate is already quite high, indicating limited room for future research.

**Strengths:**

See Pros in Review.

**Additional Feedback:**

The authors need to reconsider the necessity of conducting chicken re-identification research, as the current value of this study appears to be limited.

**Documentation:**

Yes

**Limitations:**

The limited application scenarios and research significance of this dataset are the most significant limitations.

**Opportunities For Improvement:**

-The paper should provide a compelling justification for the need to research chicken re-id. Address why this problem is significant and worth investigating, despite the absence of publicly available datasets.
-Consider comparing the value of chicken re-id research to other possible applications, such as re-id for endangered species. This could help establish a clearer value proposition and demonstrate the broader impact and importance of the research.
-Given the high success rate of chicken re-id indicated in Table 2, it is important to outline potential future research directions. Explore areas where further improvements can be made, or identify new challenges that could be addressed to advance the field.

**Relation To Prior Work:**

Yes

**Summary And Contributions:**

The article introduces "Chicks4FreeID," the first publicly available benchmark dataset for chicken re-identification, aimed at addressing the lack of annotated datasets in animal re-identification, particularly for chickens. The authors provide a baseline for re-identification using this dataset and discuss its potential for advancing animal husbandry and welfare through AI-driven tracking and management systems.

---

> ### Author Rebuttal · Authors · 2024-08-15
>
> Dear reviewer,
>
> thank you for the valuable feedback. We acknowledge that we need to better convey the benefits and applications of chicken re-ID. We also agree that a paper needs a clear value proposition and would therefore like to emphasize why chicken re-ID is indeed meaningful and worth investigating. We hope the following rebuttal will help the readers better understand the reasoning behind our work. We are also going to update our paper accordingly. Any further feedback on our rebuttal would also be greatly appreciated.
>
> # Value of conducting chicken re-ID research
> **Chicken welfare and behavioral research**
>
>
> We are convinced the welfare of all animals, including chickens, is an important topic to consider.
>
>
> To improve chicken welfare and move towards more ethical and sustainable farming, it needs to be understood how different husbandry methods impact behavior, health, and growth rates. [1][2]
>
>
> Chicken welfare assessment is increasingly focusing on individual animals rather than entire groups. [3] To effectively monitor i.e. behavior, especially when dealing with many animals in a large group, it's crucial to accurately recognize and track each individual. [3] Identifying individuals is essential not only for experiment oversight but also as a critical factor in statistical analysis.
>
>
> AI-based re-ID offers a stress-free and efficient alternative to traditional identification methods. [4] Traditional methods for re-ID, such as leg bands, wing tags, or backpacks with sensors, can cause significant stress to the animals and have been shown to negatively affect behavior, the immune system, and body weight. [5][6] This stress not only affects their well-being but also compromises the validity of the research findings. In contrast, AI-based re-ID provides a non-invasive alternative that maintains the integrity of the study while minimizing stress on the animals.
>
>
> The most recent preprint found on tackling chicken re-ID is reference [4], demonstrating that the topic is currently relevant and actively being researched. Additionally, [7] clearly recommends continuing exploring the ethological complexity of chickens in settings that are noninvasive and non-harmful, not only in commercial farming settings but also in more naturalistic settings .
>
>
> We would like to conclude by noting that reviewer AJWF positively addressed the possible application of chicken re-ID in behavior studies in their review.
>
>
> **Aiding tracking**
>
>
> Automated tracking is becoming increasingly important in chicken behavior studies [8] and is also valuable in precision livestock farming [9]. In their work, [9] state “As the frame changes, if a new chicken appears, then the old ID is dropped, and it is assigned a new ID. Hence, tracking the chicken becomes challenging due to the fact that the bird in the video may appear or disappear between the frames or there may be occlusions hiding the bird in later frames.” In this context, chicken re-identification is able to address the challenge by accurately recognizing the individual that left the frame as the same chicken upon its return.
>
> **Compliance with EU regulations**
>
>
> Chicken re-ID holds significant value in commercial farming, particularly for compliance with EU regulations [10]  that require individual identification to trace disease outbreaks.
>
> **Impact of chicken re-ID for endangered species**
>
>
> Annotating individual chickens adds value by contributing to the small pool of publicly available datasets for animal re-ID. The public datapool is actively being used in research to train species independent models. The Chicks4FreeID dataset enhances the diversity of datasets available, possibly benefitting the development of species-independent models. These models could then be applied to endangered species, thereby supporting broader conservation efforts. Chickens are birds and  all birds share one distinct feature: feathers. Our work also takes a step towards better representing birds and therefore feathers in re-ID. One prominent example of an endangered bird is the Kākāpō.
>
> # Future research
> Potential future research directions could include several exploratory areas. One possibility is to investigate which features the models prioritize during the identification process (e.g. patterns, colors, shapes, certain areas of the plumage, certain body parts) .
>
>
> OpenSet Re-ID experiments using our dataset could also be conducted, as it can certainly be adapted for open set re-ID tasks as well. For instance, researchers can exclude certain individuals from the training set and introduce them later on. To further support this task, we are willing to provide an open-set subset on Hugging Face, making it readily available for researchers interested in exploring open-set re-ID with our dataset. This aspect is further elaborated within the rebuttal for reviewer cXUF.
>
>
> It could be investigated how well our dataset can assist in re-identifying other birds where the plumage is the primary distinguishing feature. Finally, incorporating our dataset into species-independent re-ID models could be worth exploring as further explained under “Impact of chicken re-ID for endangered species”.
>
>
> Further improvements can certainly be made in the area of one shot chicken Re-ID, as our currently best experiment only achieves 56,52% top-1 accuracy (see rebuttal experiments in the attached PDF).
>
> # Rebuttal experiments, results and addressing common concerns of the reviewers
> Please see rebuttal for reviewer cXUF.
>
> # Referecnes
> added to attached PDF due to character limit

---

### Official Review · Reviewer_cXUF · 2024-07-28
**This paper proposed Chicks4FreeID, which is first dataset on chicken re-ID.**

**Rating:** 3
**Confidence:** 4
**Correctness:** The reviewer did not find fact errors…

**Review:**

This paper introduces Chicks4FreeID, the first dataset dedicated to chicken re-identification. While I appreciate the effort to extend re-identification tasks to more objects, I find the work to be incremental with no significant new insights provided. The experiment demonstrates that it is already a well-solved problem by VIT, raising questions about the necessity of a new benchmark. Additionally, I have concerns regarding the data splitting and experimental settings, which could impact the validity of the results.

**Strengths:**

Clear dataset introduction and documentation.

**Additional Feedback:**

I would be happy to raise my score if the author could address my concerns.

**Clarity:**

No validation set is introduced, which raises concerns about the generalization ability of this benchmark.

**Documentation:**

This dataset is uploaded to Hugging Face and documented well.

**Ethics:**

The author did not discuss any animal protection procedures followed during the data collection process, which is an important consideration in ethical research.

**Limitations:**

This benchmark offers incremental improvements over existing ones, which are already quite advanced. For example, in Table 2, the Vision Transformer (ViT) has already demonstrated significant success in this area.

The dataset is limited in scale, comprising only 50 unique IDs and 1270 samples. Table 1 illustrates that it contains a relatively smaller number of IDs and images compared to other animal re-identification datasets.

Additionally, this dataset focuses solely on closed-set re-identification, whereas open-set re-identification is currently more prevalent in contemporary research.

**Opportunities For Improvement:**

The short name Chicks4FreeID is puzzling—why include "Free" in the middle?

Moreover, given the recent advancements in domain transfer and unsupervised re-identification methods, I see great potential for applying domain transfer knowledge to chicken re-identification, with the possibility of achieving very high accuracy as the task appears to be relatively easy already.

**Relation To Prior Work:**

The author conducted a detailed literature review.

**Summary And Contributions:**

This dataset is the first publicly available resource for chicken re-identification. Alongside the dataset, the author provides an overview of existing animal re-identification datasets and establishes a baseline for closed-set re-identification.

---

> ### Author Rebuttal · Authors · 2024-08-15
>
> Dear reviewer,
>
> thank you for thoroughly reading and reviewing our paper. We are pleased to hear that the paper and documentation were clear and that no factual errors were identified. Furthermore, we welcome the opportunity to address your concerns. Please let us know if we have sufficiently addressed them and if you have any further questions.
>
> # Ethics
> The animals’ well-being is very important to us, and we are happy to further elaborate.
>
>
> The animals in our study are accustomed to human interaction. They were photographed in their familiar environment (barn or yard) without causing them any harm or stress. This always happened in the presence of their owners. Initial contact with the owners was made informally via phone or messaging services after they had learned about the project (from our personal and private network) and voluntarily expressed their interest. The owners were fully informed beforehand about the intent to use the data for research and subsequent publication and were given the opportunity to ask questions. Verbal consent was obtained from the owners before scheduling a date for photography.
>
>
> Given the non-invasive nature of the study and the private ownership of the animals, formal approval from an ethics committee was not sought in advance. However, we recognize the importance of ethical considerations and have since informed our Ethics Committee at Aalen University.
>
> # Experiments
> **Data Splitting**
>
>
> To avoid data leakage, we applied data augmentation only after a train-test split was established. This ensured that augmented versions of the same original image do not appear in both sets. We also did not apply any data augmentation to the test set. For a fair evaluation on all identities, the train/test split is stratified, i.e. each identity has the same fixed percentage of its cut-out crops allocated to the test set. To address your concerns, we introduced a validation set and ran the experiments anew (see attached PDF document)
>
>
>
> **Domain transfer via finetuning**
>
>
> As suggested, we conducted additional experiments (See attached PDF).
>
> # Open set re-ID
> While the current focus of our dataset is on closed-set re-identification, it can certainly be adapted for open set re-ID tasks as well. For instance, researchers can exclude certain individuals from the training set and introduce them later on. To further support this task, we are willing to provide an open set subset on Hugging Face, making it readily available for researchers interested in exploring open set re-ID with our dataset.
>
>
> Additionally, we would like to highlight that the expert ground truth labeling of all instances in the Chicks4FreeID dataset ensures reliable verification of the results in open set re-ID. This is a significant advantage over some existing open set re-ID datasets, which include unlabeled instances or rely on ground truth labels generated through clustering or other AI methods that may introduce labeling errors.
>
> # Dataset name “Chicks4FreeID”
> We are happy to answer this question. The name "Chicks4FreeID" is a playful reference to the song "Money for Nothing" by Dire Straits, which includes the phrase "chicks for free." The "Free" in the title emphasizes that our dataset is free to use ( CC BY 4.0 license) and freely available on Hugging Face. We aimed for a name that reflects both the nature of the content (Chicks), its accessibility (4Free) and purpose (ID). However, we understand that cultural differences might make the song reference less apparent.
>
> # Rebuttal experiments, results and addressing common concerns of the reviewers
> We have rerun the experiments considering the reviewers' feedback. As requested, we:
>
> + Introduced a validation split (10% of the training data, stratified random split)
> + Included instances of all visibility ratings: “best” - “good” - “bad” and therefore made it more challenging
> + Fine tuned the pretrained MegaDescriptor
> + conducted additional one-shot experiments
>
> Other parameters were left unchanged. This has resulted in the baseline now being set at 91.91% top-1 accuracy (previously 92.8%). This is indeed a good result, however, it still means that among 11 instances, at least 1 is misclassified. We argue that there is still room for improvement.
>
> It is important to note that the mAP focuses more on the quality of the probability estimates, serving as a ranking metric that assesses "how well the model assigns high probabilities to the correct chicken." This is similar to the top-5 accuracy, which, while instilling confidence in the predictions, does not fully capture the model's actual performance in terms of the number of errors made. Therefore, the top-1 accuracy must be considered.
>
> We agree that domain transfer/fine tuning is likely the right direction. We repeated the MegaDescriptor experiment, but instead of using just the pretrained embeddings, we attempted to fine tune them by unfreezing the embedding network's layers and continuing training with our dataset, using the pretrained weights as a foundation. While our results improved compared to training from scratch (Swin-L-384), they still did not surpass the performance of the ViT. This should not discourage further efforts in fine tuning /domain transfer for this problem, as this was merely our initial attempt. We believe that continued work in this area could eventually lead to successfuly solving the dataset.
>
> Furthermore, there seems to be significant room for improvement in the one shot re-ID setting with the best experiment achieving only 56,52% top-1 accuracy.
>
> It is ultimately up to the reviewers to decide if the Chicks4FreeID dataset qualifies as a benchmark dataset or should be regarded as nearly solved by our own experiments. On a final note, we kindly ask the reviewers to also evaluate our contributions besides the benchmark aspect. No matter the final decision, we appreciate your thoughtful review and are committed to improving our work based on your feedback.

---

> > ### Comment · Reviewer_cXUF · 2024-09-01
> > **Reply**
> >
> > Thank you for your detailed reply.
> >
> > A formal approval from an ethics committee is a must for most human/animal-related experiments. Though I appreciate the author's efforts, I still need to raise this concern on ethics.
> >
> > With a new validation set, VIT still achieved 91.91% top-1 accuracy, still showing the data set is saturated.
> >
> > Thus I intend to keep my rating.

---

### Author Response · Authors · 2024-08-29

Dear reviewers,

we are greatful for the detailed reviews you provided and sincerely want to thank you for the valuable feedback so far. We greatly appreciate the time and effort you have dedicated to evaluating our work.

In return, we have strived to provide thorough and thoughtful rebuttals to resolve your concerns and clarify relevant sections. As the discussion period comes to a close and we have not received any further objections, we trust our responses have been satisfactory. Should you see fit, an updated evaluation would be greatly appreciated.

---

### Decision · Program_Chairs · 2024-09-26

**Decision:**

Reject

**Comment:**

The authors present a database of pictures of chickens collected in their natural farm. The chickens have been masked and individually identified, and the authors show that the proposed algorithm can achieve high probability of identifying the correct animal. This data set does not solve a real problem in farm management. If identifying every animal is needed, tagging provides perfect precision and is cost-efficient. Computer vision solutions might miss parts of the farm and have cost and coverage issues.